

# Spatiotemporal variability of CO₂, N₂O and CH₄ fluxes from a semi-deciduous tropical forest soil in the Congo basin

Roxanne Daelman[1,2], Marijn Bauters[2], Matti Barthel[3], Emmanuel Bulonza[4], Lodewijk Lefevre[1,2], José Mbifo[5], Johan Six[3], Klaus Butterbach-Bahl[6,7], Benjamin Wolf[7], Ralf Kiese[7], Pascal Boeckx[1]

[1]Isotope Bioscience Laboratory (ISOFYS), Department of Green Chemistry and Technology, Ghent University, Ghent, Belgium
[2]Q-Forest lab, Department of Environment, Ghent University, Ghent, Belgium
[3]Department of Environmental System Science, ETH Zurich, Zurich Switzerland
[4]Ecole Régionale d'Aménagement et Gestion Intégrés des Forêts et Territoires tropicaux (ERAIFT), Kinshasa, Democratic Republic of Congo
[5]Institut National pour l'Etude et la Recherche Agronomiques (INERA), Yangambi, Democratic Republic of Congo
[6] Land-CRAFT, Department of Agroecology, Aarhus University, Aarhus, Denmark
[7]Institute of Meteorology and Climate Research, Atmospheric Environmental Research (IMK-IFU), Karlsruhe Institute of Technology, Garmisch-Partenkirchen, Germany

*Correspondence to*: Roxanne Daelman (Roxanne.Daelman@Ugent.be)

**Abstract.** Tropical forests play an important role in the greenhouse gas exchange between biosphere and atmosphere. Despite holding the second largest tropical forest globally, the Congo basin is generally understudied and ground based greenhouse gas flux data are lacking. In this study, high frequency measurements spanning of sixteen months from automated and manual soil chambers are combined, to characterize spatio-temporal variability in soil greenhouse gas fluxes from a lowland tropical forest in Yangambi, in the Congo Basin. Based on sub-daily continuous measurements, for CO₂, a total emission of $15.3 \pm 4.4$ Mg C ha⁻¹ yr⁻¹ was calculated, with highest fluxes at the start of the wetter periods and a decline in emissions during drier periods. For CH₄, the total uptake was $-3.9 \pm 5.2$ kg C ha⁻¹ yr⁻¹. Over the whole period the soil acted as a sink however sporadic emission events were also observed. For N₂O an emission of $3.6 \pm 4.1$ kg N ha⁻¹ yr⁻¹ was calculated, which is higher than most previously reported tropical forest estimates. N₂O emissions decreased substantially during drier periods and emission pulses were detected after rain events. High spatial and temporal variability was observed for both CH₄ and N₂O, but less for CO₂. Higher spatial variability was assessed by the manual compared to the automated measurements. Overall, the tropical forest soil acted as a major source for CO₂ and N₂O and a minor sink for CH₄.

## 1 Introduction

The three most important biogenic greenhouse gases (GHG) are carbon dioxide (CO₂), methane (CH₄), and nitrous oxide (N₂O). Although the increase in atmospheric concentrations of these key GHG since 1750 is unequivocally attributed to human activity (IPCC, Calvin et al., 2023), natural ecosystems also contribute significantly to atmospheric GHG budgets. Tropical forests play an important role in the GHG exchange between the biosphere and the atmosphere. Soils are the



dominant terrestrial source of $CO_2$ and emissions from tropical forest soils are generally higher than from any other vegetation type, due to high soil moisture and temperature (Raich et al., 2002), higher gross primary production (GPP) of
tropical forests compared to other forests and the larger proportion of GPP used for autotrophic respiration (Anderson-Teixeira et al., 2016). Soils can both produce and consume $CH_4$, but globally soils are the largest biotic sink for atmospheric $CH_4$. Aerobic forest soils consume more $CH_4$ than other ecosystems, but can become a source when inundated (Dalal et al., 2008; Dutaur et al., 2007). Considering all natural sources of $N_2O$, soils are the largest contributor. In particular, emissions from tropical forest soils are 50% higher than the global average for soils (Tian et al., 2020), as these tropical forests are
generally N rich systems (Brookshire et al., 2012; Hedin et al., 2009).  Taking all biogenic GHG together, the net global warming potential of tropical forest systems, suggests to be a small sink (Dalal et al., 2008), but large uncertainties remain as the spatial and temporal variability of tropical forest soil fluxes still remain poorly understood (Arias-Navarro et al., 2017; Courtois et al., 2019).

The difficulty of understanding the dynamics of soil fluxes lies in the multitude of controlling environmental factors and the complexity of the biological and physical mechanisms which lead to a high spatial and temporal variability (Courtois et al., 2018; Vargas et al., 2011). To date, the majority of soil flux measurements have been carried out using manual static chambers; a technique that is able to capture spatial variability up to a certain degree but is labour-intensive and time-consuming. In addition, measurements are generally taken on weekly to monthly time intervals, thus not covering diurnal
patterns and resulting in a lower accuracy than higher frequency measurements (Barton et al., 2015). Also, no responses to precipitation events or nocturnal measurements will be present (Courtois et al., 2019; Vargas et al., 2011). Automated soil chambers, on the other hand, allow to capture both diurnal and seasonal variability and even detect short term variations or responses to rapidly changing environmental conditions. Nevertheless, a small number of chambers reduces the spatial scale of the measurements, leading to large uncertainties in the quantification of landscape GHG fluxes (Wangari et al., 2022).
Combining automated chambers with fast box measurements can address both the spatial and temporal variability of soil GHG fluxes. However, the complexity of such setup has limited its application in remote environments such as tropical forests, particularly in Africa.

Despite being the second largest tropical forest, the Congo Basin is generally understudied (Malhi et al., 2013). Recent
studies show that the climatic conditions in the Congo Basin are already changing with an increasing length of the dry season (Jiang et al., 2019), warming (Dezfuli, 2017; Kasongo Yakusu et al., 2023) and increasing numbers of extreme warm days and nights (Aguilar et al., 2009; Chaney et al., 2014; Kasongo Yakusu et al., 2023) and increasing precipitation intensity (Kasongo Yakusu et al., 2023). Climate change projections suggest that these trends will persist in the future (Karam et al., 2022; Kendon et al., 2019). As soil moisture and temperature are main drivers of  soil GHG flus dynamics, these shifts will
directly affect the GHG budget (Ni et al. 2018). Quantifying soil fluxes and understanding the role of temperature and moisture as drivers is key to assessing the future response of this biome to climate change. So far only a handful of studies



have reported *in situ* soil flux measurements from the Congo Basin (Fig. S1 and Table S1). Studies such as Barthel et al. (2022), combine biweekly measurements from long-term observation sites, with short-term daily measurement campaigns. However, to calculate more robust GHG budgets and to understand variability, data with a higher spatio-temporal resolution

are needed.

In this study, a combination of automated and manual fast box chamber measurements were used, to quantify and understand the spatio-temporal variability of soil GHG fluxes in a semi-deciduous tropical forest in the Congo Basin. The objectives of the study are 1) to quantify annual budgets of soil $CO_2$, $CH_4$ and $N_2O$ fluxes, 2) to evaluate the role of soil

moisture and temperature as main drivers of these GHG fluxes, and 3) to analyse the spatio-temporal variability of soil GHG fluxes.

## 2 Methods

### 2.1 Study site

The study was conducted at the CongoFlux station (0°48′52.0″ N 24°30′08.9″ E) (Fig. S2) (Sibret et al., 2022), located in

Yangambi on the right bank of the Congo River, app. 100 km northwest of Kisangani, in the Tshopo Province of the Democratic Republic of Congo (DRC). The site is located in a semi-deciduous, lowland mixed-species forest with strongly weathered, sandy clay loam and poorly drained soils, dominated by Haplic Ferralsols (Sibret et al., 2022). The soil in the top 30 cm has an average bulk density (BD) of 1.18 g cm$^{-3}$, a pH-$H_2O$ of 4.0, with a clay, sand, and silt mass percentage of respectively 30 %, 68 % and 2 %. The average C content is 1.3 % and the average N content is 0.22 % (Table S2). The

region has a warm and humid climate with a bimodal rain regime where two dry seasons (Dec-Feb and June-July) alternate with two wet seasons (March-May and August-November) (Kasongo Yakusu et al., 2023). The region experiences a mean annual rainfall of 1822.19 ± 214.80 mm and has a mean annual temperature of 25.0 ± 0.30 °C (Likoko et al., 2019). Within the CongoFlux site, four permanent forest plots of 1 ha (Fig. S2) have been installed according to the RAINFOR-Gem field protocol (Marthews et al., 2013)

### 2.2 Climatic variables

Half-hourly precipitation at the site was measured by a tipping bucket (ARG100, Campbell scientific Inc., Logan, Utah, USA) installed at 56 m on the tower used for eddy covariance measurements at the study site. Air temperature was measured using temperature and relative humidity sensors (HC2S3, Campbell scientific Inc., Logan, Utah, USA) installed at 2 m height at the tower. Due to high lightning intensity at the site, power outages were frequent, resulting in several data gaps

during the measurement period. At each chamber location, two water content reflectometers were installed (CS650, Campbell Scientific Inc., Logan, Utah, USA). The sensors were installed at a depth of 5 cm and 15 cm, at a distance of



around 0.5 m from the collars to avoid disturbance of the soil in the chambers. Every minute the volumetric water content (VWC), soil temperature and electrical conductivity were logged. WFPS was then calculated following equitation 1.

$$WFPS = 100 \times VWC \times \left(1 - \frac{BD}{2.65}\right)^{-1} \quad (1)$$

Note: there is a large data gap from the end of March up to the middle of July 2023, due to technical problems with the data logger.

### 2.3 Measurements of soil $CO_2$, $CH_4$ and $N_2O$ exchange

#### 2.3.1 Automated chamber measurements

At the CongoFlux site, nine custom-made dynamic automated chambers (Karlsruhe Institute of Technology) were installed
in the 1 ha GEM plot CF1 (Fig. S2). The chambers (0.5 m x 0.5 m x 0.15 m, length, width, and height) were controlled by a central steering unit consisting of a valve-tubing system connecting chambers to two portable analysers, one measuring $CO_2$ and $CH_4$ (LI7810, LI-COR inc., Lincoln, Nebraska, USA) and the other measuring $N_2O$ (LI7820, LI-COR inc., Lincoln, Nebraska, USA) concentrations. The nine chambers were placed randomly around the control unit at a distance of maximum 20 m. For each chamber two collars were inserted 10 cm into the soil and chambers were relocated from one collar to the
other every two or three weeks. Only chamber 6 was never relocated during the measurement period due to limited cable length. The chamber headspace air was circulated from the chamber to the analysers and back to the chamber with 1/8 inch stainless steel tubing. Before the analysers sample air was dried to constant water vapour with a Nafion dryer. A portable computer was used for steering the valve system for consecutive chamber sampling, closing and opening chambers as well as data acquisition and storage.

The chambers were installed in May 2022 and were operated with a closure time of fifteen minutes per flux measurement. The flux rate was calculated using a linear fit. Measurements of $CO_2$, $CH_4$ and $N_2O$ were discarded if the R² of the $CO_2$ measurement was smaller than 0.9, and measurements of $N_2O$ were also discarded if the R² of the $N_2O$ measurement was smaller than 0.7, as these values indicated a systematic error such as imperfect closure of the chamber or a technical issue with the analysers. Fluxes with unit mass C h$^{-1}$ m$^{-2}$ for $CO_2$ and $CH_4$ and mass N h$^{-1}$ m$^{-2}$ for $N_2O$ were than calculated using
equation 2:

$$Flux = \frac{dq}{dt} \times \frac{60 \times V \times P \times M_w}{R \times T \times A \times 1000} \quad (2)$$

With $\frac{dq}{dt}$ the change in mixing ratio over time [ppb minute$^{-1}$ or ppm minute$^{-1}$] , resulting from the exponential fit, V the chamber volume [m³], P the long term average air pressure of the site [Pa], $M_w$ = 12 or $CO_2$ and $CH_4$ and $M_w$ = 28 for $N_2O$ [g mol$^{-1}$], T the average temperature during closure time [°K], A the surface area of the chamber [m²] and R the universal gas
constant [J mol$^{-1}$]. The data presented in this paper starts from the first of June 2022 up until the 26$^{th}$ of September 2023, resulting in a coverage of sixteen months and 25 209 data points for $CO_2$ and $CH_4$ and 18 635 data points for $N_2O$.



### 2.3.2 Fast box measurements

The four GEM plots on the CongoFlux site were divided into twenty-five 20 m by 20 m subplots and in each subplot, one soil chamber was installed to be measured using the fast box method. The chambers were PVC tubes with a diameter of 5.5 cm and a height of 13 cm, permanently inserted 3 cm into the soil. The four plots CF1, CF2, Mi2 and Mi5 were equipped with twenty-five chambers each, but three chambers were lost (two in Mi2 and one in Mi5) and two locations were measured incorrectly (one in CF2 and one in Mi2), leaving a total of ninety-five chambers. During a period of three weeks from August 4 to August 28, 2023, flux measurements of $CO_2$, $CH_4$ and $N_2O$ were made in these plots using two portable analysers, one measuring $CO_2$ and $CH_4$ (LI7810, LI-COR inc., Lincoln, Nebraska, USA) and one measuring $N_2O$ (LI7820, LIC-COR inc., Lincoln, Nebraska, USA). Plots CF1, CF2 and Mi5 were each measured six times during this three-week period and plot Mi2 only five times. The plots were measured at different times of the day, taking care to change the route from chamber to chamber. A closure time of two minutes was used, and the flux rate was calculated using a linear fit and equation 2.

### 3 Statistical Analysis

For the automated chambers, the average flux was calculated as the arithmetic mean of all measurements from all chambers and the average fluxes per chamber location were calculated by taking the arithmetic mean of all measurements from each chamber location separately. The coefficient of variation (CV) for each chamber location was calculated as the standard deviation (SD) of the chamber location divided by the arithmetic mean of the chamber location. The CV between chambers location was calculated as the SD of the seventeen average fluxes per chamber location, divided by the arithmetic mean of the seventeen average fluxes per chamber location. Daily fluxes were calculated by taking the arithmetic mean of all measurements from all seventeen chamber locations for each day and the CV between days was calculated as the SD divided by the arithmetic mean of all daily fluxes. Annual budgets were calculated by adding the daily fluxes for one complete year. Days without flux measurements were filled using a linear interpolation if the gap was smaller than or equal to 10 consecutive days. For each of the ninety-five fast box chambers, an average flux was calculated by taking the arithmetic mean of all measurements from one chamber during the measurement period. These fluxes were then averaged per plot, and the CV for each plot was calculated as the SD divided by the mean of the average fluxes per chamber. To investigate the effect of potential drivers of GHG fluxes, i.e., WFPS, soil temperature, air temperature and precipitation, linear mixed models were fitted using the nlme package, version 3.1-164 (Pinheiro et al., 2023), with the measurements from the automated chambers for each GHG. This was done using chamber number and collar numbers (i.e., location and position) as a random intercept and WFPS, soil temperature ($Temp_{soil}$), air temperature ($Temp_{air}$), accumulated rainfall of the past 30 minutes (Rain) and accumulated rainfall of the past 10 days ($PrevRain_{10days}$) as fixed factors. To account for the temporal correlation in the data, a first order auto-regressive model was included in the fit (CorAR1). The $CO_2$ and $N_2O$ fluxes were log-transformed and the $CH_4$ fluxes were log-transformed after adding the minimum value of the fluxes. Fixed factors were





removed from the fit if they had low t-values and removing them from the fit did not substantially increase the AIC and BIC

values of the fit. R² values were calculated according to Nakagawa et al. (2013) and partitioned following Stoffel et al. (2021). Collinearity of predictors was tested using the variance of inflation factor (VIF). Effect sizes of the fixed factors are expressed as relative changes (%) in flux per unit increase in the fixed factor and are calculated by transforming the effect size y by exp(y) - 1. Note that for $CH_4$ this relative change is calculated as a function of fluxes with the minimum flux added. Statistical difference between chamber locations for WFPS and soil temperature was tested with the Kruskal-Wallice test

(nonparametric), followed by a Wilcox test. Statistical difference between hours for diel cycles was tested with the Friedman test (nonparametric, repeated measures), followed by a Wilcox test. Results were considered significant if the p-values or Bonferroni adjusted p-values were smaller than 0.05. To look into the effect of the high sampling frequency of the set-up, a resampling procedure was carried out with the daytime measurements to simulate lower sampling frequencies with the same number of chambers by using a bootstrap with size 1000. The resampling scenarios were: a) one measurement per chamber

every month, b) one measurement per chamber every week, and c) one measurement per chamber every day. The mean, minimum, maximum, interquartile range (IQR), and normalized interquartile range (NIQR) of the resulting budgets were calculated for each scenario. The NIQR was calculated as the IQR divided by the mean value and gives an indication of the spread of the resulting budgets, compared to the mean value. More information about the resampling procedure can be found in the supplementary material. All statistical analyses were performed using R software.

**4 Results**

**4.1 Climatic Variables**

Total accumulated precipitation during the 16 month measuring period was 2181.7 mm, with highest weekly accumulated precipitation of 133.7 mm in early of September (week 36) of 2022, followed by week 44 in early November 2022 and week 16 around half of April 2023 (Fig. S3). November 2022 was the wettest month of the measurement period and December

2022, followed by May 2023 were the driest months (Table S3). June 2023 had twice the rainfall of June 2022, which made June 2023 a considerably wet month for the short dry season, while May 2023 was drier compared to May 2022. August and September 2023 were dry for the start of the long wet season, each having around half of the amount of rainfall than the same month in 2022. The average hourly air temperature was 23.8 ± 2.5°C with a hourly minimum of 17.8 °C during the night and a maximum of 32.3 °C during the day (Fig. S3).


Average WFPS at 5 cm depth was 27.8 ± 5.9 %, with an average soil temperature of 23.9 ± 0.6 °C, while average WFPS at 15 cm depth was 19.8 ± 4.1 % with an average soil temperature of 24.0 ± 0.5 °C. WFPS followed the seasonality of rainfall (Table S3) and ranged from a minimum at 5 cm and 15 cm of 11.3 % and 6.5 %, respectively, to a maximum of 69.9 % and 42 % (Fig. S4 and Table S5). Soil temperature ranged from a minimum at 5 cm and 15 cm of 21.7 °C and 21.9 °C,

respectively, to a maximum of 26.1°C and 25.9°C (Fig. S4 and Table S4). A diel cycle marked soil temperature dynamics at





both depths (Fig. S5). At both depths, there were pronounced differences in WFPS and soil temperature between chamber locations (Table S3).

### 4.2 Soil $CO_2$, $CH_4$ and $N_2O$ exchange of automated chambers

$CO_2$ emissions from all chambers during the measurement period ranged from 37.2 to 463.1 mg C m$^{-2}$ h$^{-1}$ with an arithmetic
mean of 174.5 ± 50.1 mg C m$^{-2}$ h$^{-1}$ and a median of 64.1 mg C m$^{-2}$ h$^{-1}$ (Fig. S6 a). Average emissions varied between chambers with lowest average emission of 124.4 mg C m$^{-2}$ h$^{-1}$ (chamber 8, collar 2) and highest average emission of 263.1 mg C m$^{-2}$ h$^{-1}$ (chamber 7, collar 1). Nevertheless, spatial variability over the entire measurement period appeared to be limited with a CV between chamber locations of 24 % (Table S6). The annual budgets, calculated for all 12 consecutive months ranged from 15.0 to 15.1 Mg C ha$^{-1}$ yr$^{-1}$ (Table S4).

The $CH_4$ flux during the measurement period ranged from -133.1 to 1209.0 µg C m$^{-2}$ h$^{-1}$ with an arithmetic mean of -44.6 ± 59.1 µg C m$^{-2}$ h$^{-1}$ and a median of -54. µg C m$^{-2}$ h$^{-1}$ (Fig. S6 b). Averaged over the entire measurement period, all chambers were a sink for $CH_4$ with average uptake rates ranging from -85.0 µg C m$^{-2}$ h$^{-1}$ (chamber 5, collar 1) to -19.2 µg C m$^{-2}$ h$^{-1}$ (chamber 7, collar 1). Although the forest soil was a sink, each chamber position except collar 1 of chamber 5 experienced at
least one period of high $CH_4$ emissions (Table S6). Such an emission period had a duration of a couple of hours up to three weeks. In total 9.4 % of all $CH_4$ measurements were $CH_4$ sources, with maximum per chamber location of 21 % (chamber 7, collar 1). The CV between chambers was 48 % indicating a strong spatial variability. The cumulative annual budgets ranged from an uptake of -3.7 to -4.1 kg C ha$^{-1}$ yr$^{-1}$ (Table S4).

The $N_2O$ emissions during the measurement period ranged from 2.8 to 841.5 µg N m$^{-2}$ h$^{-1}$ with an arithmetic mean of 40.9 ± 46.4 µg N m$^{-2}$ h$^{-1}$ and a median of 25.4 µg N m$^{-2}$ h$^{-1}$ (Fig. S6 c). Averaged emissions per chamber ranged from 22.9 µg N m$^{-2}$ h$^{-1}$ (chamber 8, collar 2) up to 65.4 µg N m$^{-2}$ h$^{-1}$ (chamber 7, collar 2) and the relatively low CV of 30 %, indicates that the spatial variability was limited over this extended period (Table S6). A cumulative annual budget for the $N_2O$ measurements is not possible due to the three month data gap.

For $CO_2$, a diel cycle was observed with emissions increasing from 10:00 in the morning up to 14:00 and decreasing again from 17:00 to 21:00 (Fig. S7 a). For $N_2O$ a small diel cycle was also observed with emissions increasing between 10:00 and 14:00 and decreasing again from 14:00 (Fig. S7 b). For $CH_4$ no diel cycle was observed (Fig. S7 c).

Daily fluxes were calculated by taking the arithmetic mean of all measurements from all nine chambers for each day. The mean daily flux for $CO_2$ was 174.0 ± 17.9 mg C m$^{-2}$ h$^{-1}$ (Fig. 1 a) with a CV between days of 10 % and a range from 103.5 to 227.7 mg C m$^{-1}$. The low CV of the daily fluxes and the low CV of each chamber location individually (13 % – 23 %, Table S6), indicate that the temporal variability for $CO_2$ was limited. The mean daily $CH_4$ flux was -44.7 ± 20.3 µg C m$^{-2}$ h$^{-1}$



(Fig. 1 b) with a CV of 45 % and a range from -83.1 to 72.0 C $m^{-2}$ $h^{-1}$. There were 11 days with a positive mean flux. These positive fluxes were mostly dominated by one or two chambers showing elevated emissions covering periods of several days. The CV of the daily averages, including all chambers, was relatively small compared to the CV of the individual chamber locations (23 % – 598 %, Table S6). The mean daily $N_2O$ flux was 39.3 ± 27.7 µg N $m^{-2}$ $h^{-1}$ (Fig. 1 c) with a high CV of 70 % and a range of 5.1 to 135.6 µg N $m^{-2}$ $h^{-1}$. The individual chamber locations also had high CV (56 % – 211 %, Table S6).

Reducing the sampling frequency of the measurements per chamber to one measurement every month and one measurements every week, led to GHG budgets with a NIQR of 19.1 % and 9.1 %, respectively, for $CH_4$ and 13.6 % and 7.0 %, respectively, for $N_2O$, indicating a large spread of the calculated budgets (Table S10 and Fig. S8). All three scenario resulted in relatively low NIQR values for $CO_2$. Resampling the data with one measurement per chamber per day resulted in a NIQR smaller than 3 % for all GHG.

## 4.3 Soil $CO_2$, $CH_4$ and $N_2O$ exchange of fast box measurements

The average $CO_2$ flux of all ninety-five fast box chambers was 189.4 mg C $m^{-2}$ $h^{-1}$ (Table 1), a value close to the average of the nine automated chambers during the same period, 176.3 mg C $m^{-2}$ $h^{-1}$. The CV between different plots was 9 % and the CV between different chambers was 36 %, which was higher than the CV of 26 % between the automated chambers during the month of August. For $CH_4$, the average uptake measured with the fast box method was -89.4 µg C $m^{-2}$ $h^{-1}$ with a CV of 42 %. Out of a total of 546 measurements for $CH_4$, sixteen measurements showed positive fluxes (3 %) distributed over eleven chambers in three different plots. Only one chamber had a positive flux over the entire measurement period. The average uptake rate measured with the fast box chambers was higher than the average uptake measured with the nine automated chambers in August, which was -58.8 µg C $m^{-2}$ $h^{-1}$. The CV between the plots was 14 % and the CV between the fast box chambers was higher than the CV of the automated chambers during this period (35 %). For $N_2O$, the average flux was 95.5 µg N $m^{-2}$ $h^{-1}$ with a high CV of 73 % between the chambers and 24 % between the plots, indicating a large spatial variability. The average flux measured by the fast box chambers was much higher than the flux measured by the automated chambers throughout the whole measurement period. Due to a technical issue, the automated chambers have no flux measurements for $N_2O$ during the month of August, however the average flux measured with the automated chambers in July and September 2023 is 23.9 µg N $m^{-2}$ $h^{-1}$ which is only a third of the value measured with the fast box chambers. The CV between the fast box chambers was much higher than the CV between the automated chambers in July and September (35 %).







**Figure 1: a) CO₂, b) CH₄ and c) N₂O fluxes of all nine chambers over the whole measuring period. In black are the daily median values, in grey the daily averages, in blue the 25th up to 75th quantile and the coloured points are outlier values. Outliers have a distance to the 25th or 75th quantile value that is larger than 1.5 times the interquartile distance. Each colour represents one chamber. Vertical grey lines depict days where the chambers were replaced from one collar to the other.**



**Table 1: For each of chambers an average flux is calculated using all measurements from that chamber during the measurement period. The values in this table are calculated using the average values per chamber. The average flux per plot for CO$_2$ (mg C m$^{-2}$ h$^{-1}$), CH$_4$ (µg C m$^{-2}$ h$^{-1}$) and N$_2$O (µg N m$^{-2}$ h$^{-1}$) together with the minimum and maximum value ($mean_{min}^{max}$) and the coefficient of variation (CV). Also the mean and CV is calculated for all chambers together and for the averages of the 4 plots.**

| Plot | CO$_2$ (mg C m$^{-2}$ h$^{-1}$) | CH$_4$ (µg C m$^{-2}$ h$^{-1}$) | N$_2$O (µg N m$^{-2}$ h$^{-1}$) |
|---|---|---|---|
| CF1 | $201.9_{60}^{470}$ | $-87.8_{-150}^{-22}$ | $83.8_{32}^{171}$ |
| | CV: 0.41 | CV: 0.41 | CV: 0.45 |
| CF2 | $203.2_{90}^{321}$ | $-105.2_{-167}^{-49}$ | $101.5_{32}^{241}$ |
| | CV: 0.30 | CV: 0.30 | CV: 0.53 |
| Mi2 | $166.3_{81}^{271}$ | $-90.1_{-146}^{40}$ | $70.8_{22}^{364}$ |
| | CV: 0.27 | CV: 0.46 | CV: 1.04 |
| Mi5 | $183.6_{71}^{403}$ | $-74.4_{-141}^{-2}$ | $124.2_{45}^{397}$ |
| | CV: 0.39 | CV: 0.48 | CV: 0.76 |
| All chambers | 189.4 CV: 0.36 | −89.4 CV 0.42 | 95.5 CV: 0.73 |
| All plots | 188.8 CV: 0.09 | −89.4 CV 0.14 | 95.1 CV: 0.24 |

## 4.3 Main Drivers of the GHG fluxes

For CO$_2$, all fixed factors were retained and all had a positive relationships with CO$_2$ fluxes (Fig. 2 a and b, Table S7). The fixed and random factors together explained up to 79 % of the variance of the fluxes. Similarly, only positive relationships were found between N$_2$O emissions and climatic variables (Fig. 2 c and d, Table S9). Rain was removed from the fit and all effects together explained 22 % of the variability. For CH$_4$, all effects together explained 35 % of the variability. Both rainfall and soil temperature were removed from the fit. A positive relationship with CH$_4$ was found for WFPS and a negative relationship for air temperature (Fig. 2 e and f, Table S8). The model did not fit the data of the CH$_4$ measurements well, as the residuals showed heteroscedasticity and non-normality. WFPS has the largest relative effect size within the range of variability for all three models Fig.2 b, d and f). The VIF for all predictors in all models were less than two.





**Figure 2: The relative fixed effect sizes (%) for the fixed factors and the centred and scaled fixed factors water filled pore space (WFPS, %), soil temperature (Temp$_{soil}$, °C), air temperature (Temp$_{air}$, °C), precipitation (Rain, mm per half hour) and the accumulated precipitation of the previous 10 days (PrevRain$_{10days}$, mm) for the linear mixed effect models of the three greenhouse gasses a) CO$_2$, b) CO$_2$ with centred and scaled fixed factors, c) CH$_4$, d) CH$_4$ with centred and scaled fixed factors, e) N$_2$O and f) N$_2$O with centred and scaled fixed factors. The whiskers indicate the 95% credible interval.**



## 5 Discussion

**5.1 Soil CO₂ emission**

The average of all the measurements from the automated chambers results in a carbon emission of $15.3 \pm 4.4$ Mg C ha$^{-1}$ yr$^{-1}$. The gap filled daily fluxes accumulated to an annual budget of 15.1 Mg C ha$^{-1}$ yr$^{-1}$ (Table S4). This value is higher than the previous estimate of 13.1 Mg C ha$^{-1}$ yr$^{-1}$ by Baumgartner et al. (2020) in the DRC, and more than double the amount of 6.3 Mg C ha$^{-1}$ yr$^{-1}$ estimated by Werner et al. (2007) in Kenya. Our values are close to the 17.0 and 18.3 Mg C ha$^{-1}$ yr$^{-1}$ estimated

by Tchiofo Lontsi et al. (2020) for an undisturbed forest area in Cameroon. Our study shows that this Congolese forest soil is a larger CO₂ source than observed for tropical forest soils in French Guiana, with an emissions between 9.3 and 13.6 Mg C ha$^{-1}$ yr$^{-1}$ (Courtois et al. 2018; Petitjean et al. 2019) and in Australia with emissions between 8 and 12 Mg C ha$^{-1}$ yr$^{-1}$ (Kiese et al. 2002). However, the emissions are comparable to those measured in Brazil, between 13.2 and 16.3 Mg C ha$^{-1}$ yr$^{-1}$ (Sotta et al., 2007; Sousa Neto et al., 2011) and lower than the emissions reported for forests in Panama with 19.7 Mg C ha$^{-1}$

yr$^{-1}$ (Pendall et al., 2010).

In this study, there was a low spatial variability between the automated chambers and between the fast box chambers and also a low temporal variability for CO₂ fluxes. was also observed in other lowland tropical forests in Kenya by Werner et al. (2006) and in the DRC by Baumgartner et al. (2020). This WFPS, soil and air temperature were the strongest drivers, each

explaining up to 8% of the variability in the data (Table S7). The random effect 'chamber ID', accounted for a large proportion of the variance. Both WFPS and soil temperature had a positive relationship with CO₂ flux, with WFPS having a larger relative effect size than soil temperature within their range of variability (centred and scaled factors) (Fig. 2). The positive relationship with WFPS is frequently observed in other studies, both in the Congo Basin and in other topical forests. During the early wetter period in July 2022, a first emission peak was observed with an increase of WFPS. However, there is

a slight decline in the following wet months although WFPS remains high (Fig. S9 a). The same was observed in forests in Panama, where the decrease in emissions was partly attributed to a depletion of readily available C substrates and nutrients (Cusack et al. 2023). Baumgartner et al. (2020) suggest that CO₂ emissions in lowland forests in the Congo Basin may be limited by C availability, which could explain the decline in emissions after the first pulse of microbial activity pulse at the beginning of the rainy season. A clear decrease in CO₂ emissions is also observed from the start of the drier period in mid-

November 2022. The following relatively large peaks in emissions align with the increase in WFPS due to sporadic rain events. These emission peaks following rewetting events are common and are referred to as the 'Birch effect' (Birch, 1958). With an increasing length of the dry season as expected from climate change predictions for the Congo Basin (Jiang et al., 2019), soil CO₂ emissions may decrease. However, with increasing rainfall intensity, which is also part of climate change projections (Kasongo Yakusu et al., 2023), the emission pulses after rewetting could become more frequent and severe. The

positive relationship with soil and air temperature can explain the diel cycle of the CO₂ emissions found in this study (Fig.



S5 and Fig. S7 a). Peak soil respiration is reached before the peak of soil temperature, which indicates that the diel cycle is also sustained by increasing autotrophic respiration (Savage et al. 2013; Winnick et al. 2020).

## 5.2 Soil $CH_4$ uptake

For $CH_4$, the total uptake is -3.9 kg C ha$^{-1}$ yr$^{-1}$ when using the arithmetic mean of all the measurements from the automated chambers and the annual budgets accumulate to an uptake between -3.7 and 4.1 kg C ha$^{-1}$ yr$^{-1}$ (Table S4). This uptake is close to estimates from previous studies in the DRC with an uptake of -3.5 kg C ha$^{-1}$ yr$^{-1}$ (Barthel et al., 2022), but is higher than the average uptake rate of tropical forest soils of -2.5 kg C ha$^{-1}$ yr$^{-1}$ estimated by Dutaur et al. (2007) and -3.0 kg C ha$^{-1}$ yr$^{-1}$ estimated by Dalal et al. (2008). In other African tropical forests similar uptake rates were reported, e.g. Cameroon -4.3 and -2.5 kg C ha$^{-1}$ yr$^{-1}$ (Tchiofo Lontsi et al., 2020) and -3.5 kg C ha$^{-1}$ yr$^{-1}$ (Macdonald et al., 1998) and Kenya -4.9 kg C ha$^{-1}$ yr$^{-1}$ (Werner et al. 2007). Uptake rates of -2.6 kg C ha$^{-1}$ yr$^{-1}$ were measured in Southwest China (Werner et al., 2006), -3.2 kg C ha$^{-1}$ yr$^{-1}$ in Australia (Kiese et al., 2003), -3.8 kg C ha$^{-1}$ yr$^{-1}$ in Brazil (Sousa Neto et al., 2011) and -1.1 kg C ha$^{-1}$ yr$^{-1}$ in French Guiana (Petitjean et al., 2019).

Considerable spatial variability was observed with a CV of 47 % and 42 % between the automated and the fast box chambers respectively. The average uptake measured with the fast box method was -89.4 µg C m$^{-2}$ h$^{-1}$ with a range of -230.8 to 256.99 µg C m$^{-2}$ h$^{-1}$, while during the same period, the automated chambers measured an average flux of -66.8 µg C m$^{-2}$ h$^{-1}$ with a range of -162.8 to 272.2 µg C m$^{-2}$ h$^{-1}$. This discrepancy may indicate that the nine automated chambers are not sufficient to cover the considerable spatial variability of $CH_4$ uptake, underlining the importance of a large spatial coverage. The CV of the daily averages (45 %) was relatively small compared to the CV of the individual chamber locations (Table S6), indicating that the temporal variability of the daily fluxes of all chambers together was relatively small, but that each chamber individually had a large temporal variability, mainly caused by the periods of emissions. This high spatial and temporal variability is commonly found in studies in tropical forests (Barthel et al., 2022; Castaldi et al., 2020; Werner et al., 2007). Resampling the automated chamber measurements with a lower sampling frequency of once a month or once a week, results in a large spread of possible $CH_4$ budgets (Table S10 and Fig. S8) which underlines the importance of a high sampling frequency.

The long-standing theory is that microbial methanogenesis can only occur in anoxic soils. Consequently, most weathered tropical forest soils are modelled as $CH_4$ sinks. However, sporadic $CH_4$ emissions have been measured in several studies. In the study by Werner et al. (2007), high WFPS together with crumbled soil structure, suggesting termite activity, were put forward as a possible explanation. Termite activity was also mentioned to explain $CH_4$ emissions in the study by Barthel et al. (2022). In a study in Costa-Rica by Calvo-Rodriguez et al. (2020), high methane emissions were associated with heavy precipitation events. While in the study by Castaldi et al. (2020), emissions were observed in well-drained soils and the authors suggested that the emissions may have been generated by anoxic hotspots of microbial activity within the overall



aerobic soil. In this study, no clear evidence of termite activity was found at the chamber locations. The WFPS during

emission periods ranged from 13.5 % up to 70.5 % (Table S5), so the emissions did not occur only during wetter periods (Fig S9 b). In the case that the emissions are triggered by heavy precipitation events, one would expect multiple chambers to emit during the same period, but in this study the timing of substantial $CH_4$ emissions was different for all chambers. Even though WFPS and accumulated rainfall were the strongest drivers in the model, they explained only up to 4.8 % of the variability (Table S8). Overall, this suggests that the emissions in our study are also associated with sporadic anoxic

microsites, controlled by methanogenesis, as suggested by Castaldi et al. (2020) (Angle et al., 2017; Lacroix et al., 2023; Teh et al., 2005). A positive relation between the flux and WFPS was found, which is frequently mentioned in other studies (Kiese et al. 2003; Werner et al. 2006; Werner et al. 2007; Sousa Neto et al. 2011) and a negative relationship between the flux and soil and air temperature was found, however the relationship with soil temperature was insignificant.

**5.3 $N_2O$**

The arithmetic mean of all measurements results in a budget of 3.6 kg N $yr^{-1}$ $ha^{-1}$. This value is close to the emission of 3.8 kg N $ha^{-1}$ $yr^{-1}$ measured in Kenya (Werner et al. 2007). However the value is higher than tropical average of 3 kg N $ha^{-1}$ $yr^{-1}$ estimated by Dalal et al. (2008), more than double of the emissions measured previously in DRC using static chambers, 1.6 kg N $ha^{-1}$ $yr^{-1}$ (Barthel et al., 2022) or measured in Cameroon 1.6 kg N $ha^{-1}$ $yr^{-1}$ (Iddris et al., 2020). The value is also higher than the 2.3 kg N $ha^{-1}$ $yr^{-1}$ estimated for Ghana (Castaldi et al. 2013) and the 2.9 kg N $ha^{-1}$ $yr^{-1}$ estimated for the Mayombe

Forest in Congo (Serca et al., 1994). The emissions from the CongoFlux site are also higher than several other measurements in tropical forests as the 0.5 kg N $ha^{-1}$ $yr^{-1}$ estimated in southwestern China (Werner et al., 2006), 2.4 kg N $ha^{-1}$ $yr^{-1}$ estimated in the eastern Amazonia (Verchot et al., 1999), 1.0 kg N $ha^{-1}$ $yr^{-1}$ in French Guiana (Petitjean et al., 2019) and 1.0, 4.4 and 7.5 kg N $ha^{-1}$ $yr^{-1}$ in Australia (Kiese et al. 2002; Kiese et al. 2003).

The $N_2O$ emissions have a high temporal variability, as the automated chambers have coefficients of variation between 56 % and 211 % (Table S6). Lowering the sampling frequency, resulted in large spread of possible $N_2O$ budgets (Table 10 and Fig. S8) reducing the accuracy and precision of the estimated budgets (Barton et al., 2015). Emission change significantly from year to year. The same months separated by only one year can differ in $N_2O$ emissions by a factor of four. This high temporal variability is consistent with most studies in tropical forests. In Australia two consecutive years of measurements in

the same region differed in $N_2O$ emission by a factor of seven (Kiese et al. 2002; Kiese et al. 2003). The CV between the fast box measurements is high (73 %) which supports the high spatial variability of $N_2O$ emissions (Barthel et al., 2022; Castaldi et al., 2013; Werner et al., 2007). The average flux of the fast box chambers during the three weeks in August 2023 is twice as high as the average flux measured by the nine automated chambers during the entire measurement period and three time as high as the average flux measured by the automated chambers in July and September 2023. The large discrepancy

between the measured fluxes from automated and fast box measurements together with the low CV between the automated



chambers, but the high CV between the fast box chambers, may indicate that the nine automated chambers are not able to capture to full spatial heterogeneity of the $N_2O$ emissions.

WFPS is the strongest driver (Table S9) and has the largest relative effect size within its range of variability compared to other predictors (Fig. 2). The positive relationships fitted by the linear mixed model are found in several studies, confirming

higher emissions during wet season, compared to dry seasons (Iddris et al., 2020; Werner et al., 2007). Shortly after rain events, $N_2O$ emissions increase rapidly and then slowly decrease again with decreasing WFPS (Fig. S9 c). From January 2023, with the onset of the drier months, the high fluxes and peaked responses to increasing WFPS seem to disappear. This phenomenon was also observed by Kiese et al. (2003) in Australia, where it could have been associated with significant changes in the composition of the microbial community. With the onset of the early wet season around June and July, the

emissions do not increase again. However, the fast box flux in August is almost the same as the high average flux measured by the automated chambers around the same period in the previous year (June and July 2022). The large difference could therefore also be the result of altered conditions  at the chamber locations due to the long deployment of the automated chambers on the same location. The relatively low marginal $R^2$ of the fit suggests that there are other main drivers responsible for the large temporal variability in $N_2O$ emissions.

**6 Conclusions**

Overall, our observations confirm that tropical forest soils are a major source of $CO_2$ and $N_2O$ and a sink for $CH_4$. With an emission of 15.1 Mg C ha$^{-1}$ yr$^{-1}$, this study identifies the soil of this study area in the Congo Basin as a larger $CO_2$ source than most tropical forest soils previously reported in the literature. For $N_2O$, the emissions at the CongoFlux site were also higher than most emissions previously measured at tropical sites. With a global warming potential 265 times greater than

$CO_2$, these high $N_2O$ emissions should be taken into account in the GHG budget of tropical forests. Large spatial and temporal variability was found for the $CH_4$ and $N_2O$ fluxes, highlighting the importance of large spatial coverage and particularly high temporal sampling frequency when measuring soil fluxes. Lowering the sampling frequency of the measurements leads to a decline in precisions of the estimated budgets. Soil and air temperature are positively related to $CO_2$ and $N_2O$ emissions. Therefore, an increasing number of warm days and nights and general warmer weather conditions could

lead to overall higher $CO_2$ and $N_2O$ emissions from soils. $CO_2$ and $N_2O$ emissions are lowest during drier periods, therefore, the increasing length of dry season could offset the effect of the increasing emissions with increasing temperatures. However, there are clear spikes in $N_2O$ emissions after rainfall events and higher $CO_2$ emissions after the onset of the rainy season, and with an increasing precipitation intensity, this could lead to higher and more frequent emission peaks for both $CO_2$ and $N_2O$, which in turn would increase the total GHG emissions from tropical forest soils. An increasing precipitation

intensity could also lead to less uptake of $CH_4$, lowering the net uptake of -3.9 kg C ha$^{-1}$ yr$^{-1}$ found in this study, but with the lower global warming potential of $CH_4$, this will only have a small influence on the total GHG emissions from forest soil.



The net effect of climate change on the GHG budget of this ecosystem is still hard to predict and should be further investigated with appropriate warming and controlled rainfall studies.

## 400 Code availability

### Data availability

The core datasets generated during the current study have been deposited in the Zenodo repository [10.5281/zenodo.12200453] and are also available from the corresponding author upon request.

### Supplement link

## 405 Author contribution

R. Daelman, M. Bauters and P. Boeckx conceived the study. R. Daelman, E. Bulonza, L. Lefevre., D.E., F.K., J. Mbifo., H.F., M.B., L.C. and H.V. conducted the fieldwork. K. Butterbach-Bahl, R. Kiese and B. Wolf designed the automated chamber setup and the code to calculate the fluxes. R. Daelman did the statistical data analysis. All co-authors substantively revised the manuscript.

## 410 Competing interests

The authors declare that they have no conflict of interest.

### Acknowledgements

The authors thank the CongoFlux team (L.L., D.E., F.K, H.F., J.M.) for assisting in the installation and for the maintenance of the chamber set up and thank master students L.C. and H.V. for assisting with the manual chamber measurements. We
thank the local community in Yangambi and all Congolese friends who drove us, cooked for us and helped us navigate the forest. We further acknowledge the support by the Institut National des Etudes et Recherches Agronomique (INERA) during our fieldwork in the Yangambi Biosphere Reserve.



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
