# Peer review of "Spatiotemporal variability of CO2, N2O and CH4 fluxes from a semi-deciduous tropical forest soil in the Congo basin"

_EGUsphere, 2024_

## Author Comment (AC1)

**Answer to Referee1: Dianming Wu: comment on EGUsphere-2024-2346**

*In this work, Daelman et al. offer a critical examination of greenhouse gas fluxes from a semideciduous tropical forest soil in the Congo Basin, an area underrepresented in the scientific literature. Utilizing a combination of automated and manual soil chamber measurements, the research provides a detailed analysis of the spatiotemporal variability of $CO_2$, $CH_4$, and $N_2O$ emissions, revealing the forest soil as a significant source of $CO_2$ and $N_2O$ and a minor sink for $CH_4$. The findings are pivotal in elucidating the nitrogen and carbon cycles within tropical forest soils, as well as in assessing the ecosystem's vulnerability to climate change.*

We thank the reviewer very much for this positive feedback.

*However, this study has the following issues:*

1. *The Congo Basin is a vast and diverse region. How do you ensure that the data collected from the specific study site in Yangambi is representative of the broader Congo Basin's tropical forest soils?*

   Thank you for your question. The Congo Basin is indeed a vast and diverse region with different climate, soil types, different forest compositions, rich biodiversity, and different forest types in general. The results of this study are therefore not representative of the whole Congo Basin. However, the CongoFlux site is situated in a lowland mixed species forest, identified as semi-deciduous with patches of evergreen forest. According to (Shapiro et al., 2021), semi-deciduous rainforest covers around 104 330 000 ha of the Congo Basin and a combination of evergreen and semi-deciduous forest covers a total area of 18 000 000 ha. In terms of vegetation, the CongoFlux site therefore represents about 33% of the entire Congo Basin, assuming 3.6 million square kilometer total size. Moreover, lowland semi-deciduous forests as found at our site represent 91% of all tropical forest types in the Congo Basin. The main soil type at our research site is Ferralsols. According to (Baert et al., 2009) Ferralsols are the dominant soil type in the DRC, which contains most of the tropical forest of the Congo Basin. We are therefore confident that our site is well-suited to represent a significant part of the tropical forest realm in the Congo Basin.

   The Congo basin in general lacks in situ-data and comparisons with the data that is available for soil fluxes show that there is quite a diversity of measurement techniques and results. With the combination of automated and fast box chamber we tried to tackle the problems of previous studies and therefore provide a more robust estimate. Although our site is representative for a large area within the Congo Basin, this estimate will never cover the entire extend of the Congo Basin. However, it is a starting point for further investigation and a benchmark for model output.

2. *You mention the use of both automated and manual soil chambers. Could you elaborate on how the data from these two different methods were integrated, and whether any corrections or normalizations were applied to ensure consistency in the dataset?*

No corrections or normalizations were applied to either dataset. We believe that the methods are comparable due to the fact that the processing steps of the two methods are the same. When comparing the datasets, we take into account that the manual chambers are only measured during a limited period of time. The same time periods are selected to avoid comparing measurements in different meteorological situations. In the article, we will mention more clearly that these overlapping periods are selected for comparison.

3. *The manuscript notes sporadic $CH_4$ emission events. What are the potential ecological or environmental triggers for these events, and how were they identified in your study?*

Thank you for pointing out that this was not entirely clear. Line 328-342 says that several papers have suggested either heavy precipitation, termite activity or anoxic hotspots of microbial activity within the overall aerobic soil were put forward as possible triggers for these sporadic $CH_4$ emissions. In this study no clear evidence of termite activity was found at the chamber locations and the emissions did not occur only during wetter periods. Overall, this suggests that the emissions in our study are also associated with sporadically occurring anoxic microsites, dominated by methanogenesis.

4. *The study identifies water-filled pore space (WFPS) as a significant driver of $N_2O$ emissions. Have you investigated other potential drivers, such as soil pH, nutrient availability, or microbial community composition, which could also influence $N_2O$ emissions?*

No, we did not investigate other potential drivers apart from WFPS in this study. We do not have information on soil pH, nutrient availability of microbial community composition for all chambers separately. However, we state in line 379-380 that the low marginal $R^2$ of the fit suggests that there are indeed also other main drivers behind our results found in the study. We were primarily interested in capturing the high spatiotemporal variability using automated chambers to obtain a robust estimate for $N_2O$ flux magnitude and variability. Sampling for the mentioned other drivers even only at a weekly frequency was not feasible and not trivial. Problems such as sample preservation hinder sensitive analysis such as microbial composition. With continued developments in the field (e.g. DNA preservation solutions) and improved infrastructure we hope to implement these parameters in future studies.

5. *The high temporal variability of $N_2O$ emissions is noted. Could the authors provide insights into the seasonal patterns and inter-annual variability observed in the study, and how this variability might be linked to climate drivers?*

In terms of seasonal variability and the drivers, we mention in the paragraph starting on Line 368 that we see higher emissions in the wet compared to the dry season. We also see higher emissions after rain events, but these seem to disappear with the onset of the dryer season. Regarding the intra-annual variability, we mention that there is a large variability between consecutive years in the automated chamber measurements as we see lower fluxes during the onset of the second wet season around June and July than in the first wet season. However, during the second wet season, higher fluxes are measured by the manual chambers, which could indicate that the soil conditions have changed, triggering the lower fluxes in the manual chambers

> LINE 368 -372: "WFPS is the strongest driver (Table S9) and has the largest relative effect size within its range of variability compared to other predictors (Fig. 2). The positive relationships fitted by the linear mixed model are found in several studies, confirming higher emissions during wet season, compared to dry seasons (Iddris et al., 2020; Werner et al., 2007). Shortly after rain events, $N_2O$ emissions increase rapidly and then slowly decrease again with decreasing WFPS (Fig. S9 c). From January 2023, with the onset of the drier months, the high fluxes and peaked responses to increasing WFPS seem to disappear."

> LINE 357: "Emissions change significantly from year to year. The same months separated by only one year can differ in $N_2O$ emissions by a factor of four."

> LINE 374 – 378: "With the onset of the early wet season around June and July, the emissions do not increase again. However, the fast box flux in August is almost the same as the high average flux measured by the automated chambers around the same period in the previous year (June and July 2022). The large difference could therefore also be the result of altered conditions at the chamber locations due to the long deployment of the automated chambers on the same location."

6. *What is the detection limit of the flux /GC? Are these real negative fluxes? please state detection limits - important when claiming negative fluxes.*

Thank you for noticing that the detection limits of the measurement instruments are not mentioned in the article. The negative $CH_4$ fluxes mentioned in the study are real in the sense that negative fluxes are generally expected for methane and $CH_4$ uptake has been measured before in other studies in similar areas, for example Barthel et al., 2022. The analyzers used in this study measure at a frequency of 1 Hertz. The linear fits are therefore performed on many datapoints which is in contrast to the static chamber method, analyzed with a GC, where linear fits are performed on 4 data points only. Therefore, we assume that the detection limits in our case are less important than when using the static chamber method linked to GC. The detection limit against background ambient concentrations of the set-up is not measured by ourself. We do know the precision of the analysers given by the company and this information is added in the supplementary material:

**Precision (1σ) of the analysers used in the set-up**

**CO₂ Measurements**
3.5 ppm at 400 ppm with 1 second averaging
**CH₄ Measurements**
0.60 ppb at 2 ppm with 1 second averaging
**N₂O Measurements**
0.40 ppb at 330 ppb with 1 second averaging

From this information we could calculate the detection limit of the flux by using 2 times this standard deviation divided by the closure time of 15 minutes as dq/dt in equation 1 from the manuscript. This results in a detection limit for $CO_2$ equal to 2.0 mg C m$^{-2}$ h$^{-1}$, for $CH_4$ equal to 0.3 µg C m$^{-2}$ h$^{-1}$ and for $N_2O$ equal to 0.5 µg N m$^{-2}$ h$^{-1}$. We also added these values to the Supplementary Material.

7. *L120, give a reference for the equation used.*

Thanks for noticing that the use of this formula was not clear. The formula is a combination between the ideal gas law and additional scaling variables to arrive at the units we work with in this article. We reformulated the equation so that it is more clear how the equation is used.

The equation is now written as:

$$Flux = \frac{dq}{dt} \times \frac{V \times P \times M_w}{R \times T} \times \frac{60}{A \times 1000} \quad (2)$$

8. *2.3.2: should provide detailed information about the experimental design for manual chamber measurements, which is crucial for understanding the methodology and interpreting the results. For example, clarify the time of day when measurements are taken and whether these times are consistent across all measurements or vary according to a specific schedule.*

The section about the fast box measurement indeed missed some crucial information for the reader to understand the method we applied correctly.

The measurements were only taken during daytime, between 08:00 and 18:00 and the route through all plots was changed every day such that the measurement timing of each chamber was different for every measurement.

We changed various line in the article to make this more clear:

> LINE 133: " During a period of three weeks from August 4 to August 28, 2023, flux measurements of $CO_2$, $CH_4$ and $N_2O$ were made during daytime (between 08:00 – 18:00) in these plots using two portable analysers …."

LINE 136: ". Two plots were measured per day with on average 5 minutes between consecutive chamber measurements."

LINE 137: "To avoid consistently measuring the same chamber on the same time of day, the order in which the plots were measured was alternated and the route from chamber to chamber within one plot was changed every session."

**References**

Baert, G., Van Ranst, E., Ngongo, M., Kasongo, E., Verdoodt, A., Mujinya, B. B., and Mukalay, J.: Guide des sols en République Démocratique du Congo, tome II: description et données physico-chimiques de profils types, 2009.

Barthel, M., Bauters, M., Baumgartner, S., Drake, T. W., Bey, N. M., Bush, G., Boeckx, P., Botefa, C. I., Dériaz, N., Ekamba, G. L., Gallarotti, N., Mbayu, F. M., Mugula, J. K., Makelele, I. A., Mbongo, C. E., Mohn, J., Mandea, J. Z., Mpambi, D. M., Ntaboba, L. C., Rukeza, M. B., Spencer, R. G. M., Summerauer, L., Vanlauwe, B., Van Oost, K., Wolf, B., and Six, J.: Low N2O and variable CH4 fluxes from tropical forest soils of the Congo Basin, Nature Communications, 13, https://doi.org/10.1038/s41467-022-27978-6, 2022.

Shapiro, A. C., Grantham, H. S., Aguilar-Amuchastegui, N., Murray, N. J., Gond, V., Bonfils, D., and Rickenbach, O.: Forest condition in the Congo Basin for the assessment of ecosystem conservation status, Ecological Indicators, 122, 107268, https://doi.org/10.1016/j.ecolind.2020.107268, 2021.

---

## Author Comment (AC2)

**Answer to Referee2: comment on EGUsphere-2024-2346**

*Forests are highly relevant to global greenhouse gas (GHG) budgets. The Congo equatorial forest is one of the forests with little knowledge of GHG fluxes. Daelman et al. presents a study on fluxes in the Congo Basin using dynamic chamber techniques. The flux numbers are robust, with results close to those obtained in other tropical forests.*

We thank the reviewer for this positive feedback.

*However, some points require further clarification or discussion:*

1. *What are the intervals between samples for both types of chambers, the average number of measurements for the daytime and nighttime periods?*

   Thank you for pointing out that this information was not clear in the manuscript. For the automated chambers, one out of nine chambers was closed every 17 minutes, resulting in one data point every 17 minutes, with almost the same number of day and night measurements due to the continued operation of the chambers. In lines 115-116, we have rephrased this information:

   > LINES 115-116: " The chambers were installed in May 2022 and were operated with a closure time of fifteen minutes per flux measurement, followed by two minutes of purging with ambient air, resulting in one datapoint every 17 minutes"

   For the fast box measurements we added that only daytime measurements were available and that there was an average of 5 minutes between consecutive measurements, which was the time that it took to walk from one chamber to the next one.

   > LINE 133: " During a period of three weeks from August 4 to August 28, 2023, flux measurements of $CO_2$, $CH_4$ and $N_2O$ were made during daytime (between 08:00 – 18:00) in these plots using two portable analysers ...."

   > LINE 136: ". Two plots were measured per day with an average of 5 minutes between consecutive chamber measurements."

2. *It is not clear in the text that the soil parameters shown in Table S2 were obtained only for the CongoFlux climate site and not for the other points of the experiment (CF1, CF2, Mi2 and Mi5). How representative are these measurements for the remaining points?*

   The article did not mentioned where the soil property data was measured. We have added to the caption of Table 2 that these measurements were made in the CF1 plot.
   We do not have information on nutrient availability or soil N for all chambers separately. The CF1, CF2 and Mi5 plots are all dominated by Haplic Ferralsols and have a clay content of around 30% to 40% , while the area of Mi2, which is the plot furthest away from the others, situated slightly lower, is dominated by both Haplic and Xanthic Ferralsols and has a slightly smaller clay content, between 20% and 30%. Mi2 is therefore slightly different from the other plots, but all soils are Ferralsols, kaolinitic, acidic with a pH in water less than 4.5, poor in organic carbon and in exchangeable cations. Therefore, only one set of soil parameters is included in the article. To give the reader more information about the

homogeneity of the soils, we have included the following soil map in the Supplementary Material (Figure S10) and added a reference to a soil map, i.e.: "Gilson, P., Van Wambeke, A. and Gutzweiler, R.: Carte des Sols et de la Végétation du Congo Belge et du Ruanda-Urundi, 6: Yangambi, planchette 2: Yangambi. Notice explicative. INEAC, Bruxelles, 1956".

[Figure]

*Figure S10: Soil map of Yangambi with the CongoFlux climate site (0°48'52.0" N 24°30'08.9" E) in the black dotted line tower and the squares indication the locations of the 4 sampling plots (CF1, CF2, Mi2 and Mi5). Source for the map is: Gibson et al., 1956*

3. *Regarding fast boxes, is there any reference to their use in an experiment like the one presented or were they designed by the authors?*

The previous use of this method was not yet mentioned in the article. Thank you for pointing this out. The fast measurements with the portable analyzer is briefly described in Hensen et al., (2013) and used in Bureau et al., (2017) and Wangari et al., (2022). We have added two references in the article.

> LINE 128: The four GEM plots on the CongoFlux site were divided into twenty-five subplots of 20 m by 20 m and in each subplot, one soil chamber was installed in March 2023 to be measured using the fast box method (Hensen et al., 2013; Wangari et al., 2022).

4. *Was there any place where the automatic chambers and fast box were placed close together in the same sampling location to evaluate the performances?*

Thank you for bringing this to our attention. The automated chambers were located to the west, just outside the CF1 plot, so the fast box chambers of the CF1 plot were closest to the automated chambers. However, even the closest fast box chambers were already between 10 and 60 meters away from the automated set up. It is expected that $CH_4$ and $N_2O$ can already vary a lot between sites within only several meters. Selecting the fast box chambers located at the CF1 plot, closest to the automated chambers, we can make the following comparison for $CO_2$, $CH_4$ and $N_2O$. The fast box measurements from the CF1 plot for $CO_2$ are quite comparable with those of the automated chambers over the same time period. The spread for the fast box method is a bit larger, which is to be expected since we have more locations and smaller chambers. In both the automated and the fast box measurements, positive fluxes for $CH_4$ are present, but the measurements are dominated by negative fluxes. Especially in the second half of august the fast box fluxes tend to be more negative than the fluxes measured with the automated chambers. For $N_2O$ no real comparison can be made due to the malfunction of the $N_2O$ analyzer of the automated soil chamber set-up. Therefore the measurements of July and September from the automated chambers are compared to the measurements of August of the fast box chambers. Our results show that the fast box measurements are generally higher than the automated measurements. The mismatch of dates could lead to a discrepancy in the averages, especially because we see a slight increase in flux for some automated chambers in early August and then a decrease again at the end of August, which could indicate a period of higher fluxes that is missed here. Line 376 notes that this discrepancy could also be due to altered soil conditions  at the automated chamber locations due to the long term deployment on the same location.
We will add figures to the supplementary material which illustrate the above points with a small explanatory text to give the reader insight into the performance of the two methods.

> "To evaluate the two methods used in this study, i.e. fast box and automated chamber method, a comparison can be made between the measurements of the automated chambers and the measurements of the fast box chambers closest located to the automated chambers (plot CF1), during the overlapping time period. The fast box measurements from the CF1 plot for $CO_2$ are quite comparable with those of the automated chambers over the same time period. The spread for the fast box method is larger, which is to be expected since there are more locations of the fast box chambers and the chambers are smaller in size. In both the automated and the fast box measurements, positive fluxes for $CH_4$ are present, but the measurements are dominated by negative fluxes. Especially in the second half of august the fast box fluxes tend to be more negative than the fluxes measured with the automated chambers. For $N_2O$ no real comparison can be made due to the malfunction of the $N_2O$ analyzer of the automated soil chamber set-up. Therefore the measurements of July and September from the automated chambers are compared to the measurements of August of the fast box chambers. Our results

show that the fast box measurements are generally higher than the automated measurements. The mismatch of dates could lead to a discrepancy in the averages, especially because we see a slight increase in flux for some automated chambers in early August and then a decrease again at the end of August, which could indicate a period of higher fluxes that is missed here. This discrepancy could also be due to altered soil conditions at the automated chamber locations due to the long term deployment on the same location."

[Figure]

Figure 1: Flux measurements for $CO_2$, $CH_4$ and $N_2O$, with in red the measurements from the fast box chambers in plot CF1 and in black the measurements from all automated chambers.

5. *Considering the occurrence of precipitation almost every day (Fig. S3), what is the strategy for measurements with these events?*

Rainfall was indeed a frequent event. During rain events, the automated chambers continued to measure as normal, without interruption. Originally, a longer chamber closure time was planned (45 minutes), with several chambers closing at the same time. A tipping bucket was installed in order to open all chambers during heavy rain events to ensure that the soil in the closed chambers did not miss out on an entire rain event and receive as much precipitation as the surrounding soil. However, we quickly shortened the closure time to 15 minutes, which reduced the chance of missing out on the whole rain event and only one chamber was closed at a time, which meant that only one chamber missed out on a part of the rain events. Therefore, we can assume that continuing the measurements during the rain events would not bias the data.

Manual measurements were continued during drizzle and light rain events, but were interrupted during two heavy rain events. Measurements were resumed as soon as possible after the rain event.

6. *In tab. 1, inform that the data refers to fast boxes.*

Thank you for reading the captions thoroughly. We have added in each caption if the metrics/measurements are from either the automated or the fast box chambers.

7. *Check if the $CH_4$ fluxes reported in lines 201 and 316 are correct, with the aforementioned tables.*

Thank you for checking the values in detail. The values on line 201 are correct. The range -133.1 to 1209.0 µg C $m^{-2}$ $h^{-1}$ can be found in Table S6, with chamber 4, collar 2 having the maximum value and chamber 5 collar one having the minimum value. The arithmetic mean in Line 201( -44.6 ± 59.1 µg C $m^{-2}$ $h^{-1}$ ) is the mean of all measurements over all chambers and all collars. This mean value is not the same as the mean mentioned in the last row of Table S6 (-45.2 ± 21.8 µg C $m^{-2}$ $h^{-1}$). This last value is the mean and standard deviation of all collar means.
The mean value of the fast box measurements in line 315 (-89.4 µg C $m^{-2}$ $h^{-1}$ ) is correct and can be found in Table 1. The range -230.8 to 256.99 µg C $m^{-2}$ $h^{-1}$ is the range of all measurements and not the average of the measurements per chambers, which are shown in Table 1.

The mean and range values of the automated chambers in line 316 (-66.8 µg C $m^{-2}$ $h^{-1}$ with a range of -162.8 to 272.2 µg C $m^{-2}$ $h^{-1}$) are calculated using the same time period as the fast box measurements are carried out and are therefore not the same as mentioned in line 201.

We added now small indications in the article to make these differences more clear. We added the reference to the table for the ranges and added the words 'calculated with all measurements' in the text to make clear that this is a different value from the means in Table S6, .

In the caption of Table S6, it is written that the "all chambers" row is calculated with the mean values per chamber, and not with all measurements combined.

> LINE 194-195: $CO_2$ emissions from all chambers during the measurement period ranged from 37.2 to 463.1 mg C $m^{-2}$ $h^{-1}$ (Table S6) with an arithmetic mean, calculated with all measurements, of 174.5 ± 50.1 mg C $m^{-2}$ $h^{-1}$ and a median of 64.1 mg C $m^{-2}$ $h^{-1}$ (Fig. S6 a).

> LINE 2021-202: The $CH_4$ flux during the measurement period ranged from -133.1 to 1209.0 µg C $m^{-2}$ $h^{-1}$ (Table S6) with an arithmetic mean, calculated with all measurements, of -44.6 ± 59.1 µg C $m^{-2}$ $h^{-1}$ and a median of -54. µg C $m^{-2}$ $h^{-1}$ (Fig. S6 b).

> LINE 210-211: The $N_2O$ emissions during the measurement period ranged from 2.8 to 841.5 µg N $m^{-2}$ $h^{-1}$ (Table S6) with an arithmetic mean, calculated with all measurements, of 40.9 ± 46.4 µg N $m^{-2}$ $h^{-1}$ and a median of 25.4 µg N $m^{-2}$ $h^{-1}$ (Fig. S6 c).

> LINE 315-316: The average uptake measured with the fast box method was -89.4 µg C $m^{-2}$ $h^{-1}$ with a range of all measurements separately of -230.8 to 256.99 µg C $m^{-2}$ $h^{-1}$, while during the same period, the automated chambers measured an average ...

8. *Still in relation to the fast boxes, when evaluating the performance of the chambers, it should be taken into account that with their small area, an increase in the variability of the fluxes was expected due to edge effects, while a larger area of the automatic chamber would reduce this influence on the fluxes.*

Thank you for bringing this up. The larger number of smaller chambers would indeed lead to an increase in variability compared to a smaller amount of larger chambers. In this study we had to take the practicality and the feasibility of the method into account. We chose for the small and easy to handle chambers because the vegetation in the plots was dense, the trails were not easy to walk and the distances between plots were sometimes long. Larger chambers, to match the size of the automated chambers would have been very difficult to manage with the small field team we had.

However the effect of installing the edges into the soil, can be ignored in this study. The manual chambers were installed in March 2023. The chamber measurements used in this study were taken in August 2023 and so the effects of installation can be discarded. This information was not previously included in the article, we have now added this information in Line 128-129:

> LINE 128: "The four GEM plots on the CongoFlux site were divided into twenty-five 20 m by 20 m subplots and in each subplot, one soil chamber was installed in March 2023 to be measured using the fast box method."

**References**

Bureau, J., Grossel, A., Loubet, B., Laville, P., Massad, R., Haas, E., Butterbach-Bahl, K., Guimbaud, C., and Hénault, C.: Evaluation of new flux attribution methods for mapping N2O emissions at the landscape scale, Agriculture, Ecosystems & Environment, 247, 9–22, https://doi.org/10.1016/j.agee.2017.06.012, 2017.

Hensen, A., Skiba, U., and Famulari, D.: Low cost and state of the art methods to measure nitrous oxide emissions, Environ. Res. Lett., 8, 025022, https://doi.org/10.1088/1748-9326/8/2/025022, 2013.

Wangari, E. G, Mwanake, R. M., Kraus, D., Werner, C., Gettel, G. M., Kiese, R., Breuer, L., Butterbach-Bahl, K., and Houska, T.: Number of Chamber Measurement Locations for Accurate Quantification of Landscape-Scale Greenhouse Gas Fluxes: Importance of Land Use, Seasonality, and Greenhouse Gas Type, Journal of Geophysical Research: Biogeosciences, 127, e2022JG006901, https://doi.org/10.1029/2022JG006901, 2022.

---

## Author Comment (AC3)

**Answer to Referee3: comment on EGUsphere-2024-2346**

*This work presented GHG fluxes from tropical forest in Congo Basin where there are large gaps of knowledge. By combining automatic chamber and manual chamber methods, both temporal and spatial variability are examined. Soils of this study site was shown to be large sources of $CO_2$ and $N_2O$ with high spatiotemporal variations, highlighting the importance of extensive research.*

We thank the reviewer very much for this positive feedback

*Some more questions are listed below:*

1. *Please provide more details in 2.3.1 if the automatic chamber is opaque or not, this determine the measured $CO_2$ fluxes are NEE or respiration.*

   Thank you for noticing; this is important information that was not in the article. We added in Line 105 that the chambers were opaque.

   > LINE 105: "The opaque chambers (0.5 m x 0.5 m x 0.15 m, length, width, and height) were controlled by a central steering unit consisting of a valve-tubing system connecting chambers to two portable analyzers … "

2. *Line 130 described the manual chamber was permanently installed into soil, when they are installed and do you take into account any effect caused by the installation? Did you do quality control of fast box fluxes measurement as for automatic chamber? What's the percentage of bad quality measurements that are discarded?*

   The section about the fast box measurement was indeed missing some crucial information for the reader to understand the method we applied correctly.

   The installation of the manual chambers took place in March 2023. The chamber measurements used in this study were carried out in August 2023 and so the effects of installation can be discarded. We have now added this information now in Line 128-129, stating:

   > LINE 128: The four GEM plots on the CongoFlux site were divided into twenty-five subplots of 20 m by 20 m and in each subplot, one soil chamber was installed in March 2023 to be measured using the fast box method (Hensen et al., 2013; Wangari et al., 2022).

   The quality control performed for the fast box measurements was the same as for the automated chambers, except that measurements with low $R^2$, were also individually checked for their quality. Only 5 data points were removed. The fluxes were measured manually with an analyser where the increase in greenhouse gas concentration could be seen visually on the user interface of the analyser. Bad seals or other problems were easily detected and the measurement was then restarted. Therefore, this small amount of poor quality data is as expected. We added this information to Line 138:

   > LINE 138: The quality control of these fluxes was similar to that of the automated fluxes, with the addition that fluxes with a low $R^2$ were also individually checked for their quality.

3. *It's not clear where the environmental variables are measured, is the "each chamber location" in Line 95 refers to which kind of chamber or both?*

This was indeed not clear. We have added a sentence stating that these variables, nl VWC and soil temperature, are measured at the automated chamber locations and not at the fast box locations.

> LINE 95: At each automated chamber location, two water content reflectometers were installed ...

4. *Since automatic chambers are at plot CF1, could you compare their results with fast box result in same plot to have an idea of the performance of the two methods?*

Thank you for bringing this to our attention. The automated chambers were located to the west, just outside the CF1 plot, so the fast box chambers of the CF1 plot were closest to the automated chambers. However, even the closest fast box chambers were already between 10 and 60 meters away from the automated set up. It is expected that $CH_4$ and $N_2O$ can already vary a lot between sites within only several meters. Selecting the fast box chambers located at the CF1 plot, closest to the automated chambers, we can make the following comparison for $CO_2$, $CH_4$ and $N_2O$. The fast box measurements from the CF1 plot for $CO_2$ are quite comparable with those of the automated chambers over the same time period. The spread for the fast box method is a bit larger, which is to be expected since we have more locations and smaller chambers. In both the automated and the fast box measurements, positive fluxes for $CH_4$ are present, but the measurements are dominated by negative fluxes. Especially in the second half of august the fast box fluxes tend to be more negative than the fluxes measured with the automated chambers. For $N_2O$ no real comparison can be made due to the malfunction of the $N_2O$ analyzer of the automated soil chamber set-up. Therefore the measurements of July and September from the automated chambers are compared to the measurements of August of the fast box chambers. Our results show that the fast box measurements are generally higher than the automated measurements. The mismatch of dates could lead to a discrepancy in the averages, especially because we see a slight increase in flux for some automated chambers in early August and then a decrease again at the end of August, which could indicate a period of higher fluxes that is missed here. Line 376 notes that this discrepancy could also be due to altered soil conditions at the automated chamber locations due to the long term deployment on the same location.
We will add figures to the supplementary material which illustrate the above points with a small explanatory text to give the reader insight into the performance of the two methods.

> "To evaluate the two methods used in this study, i.e. fast box and automated chamber method, a comparison can be made between the measurements of the automated chambers and the measurements of the fast box chambers closest located to the automated chambers (plot CF1), during the overlapping time period. The fast box measurements from the CF1 plot for $CO_2$ are quite comparable with those of the automated chambers over the same time period. The spread for the fast

box method is larger, which is to be expected since there are more locations of the fast box chambers and the chambers are smaller in size. In both the automated and the fast box measurements, positive fluxes for $CH_4$ are present, but the measurements are dominated by negative fluxes. Especially in the second half of august the fast box fluxes tend to be more negative than the fluxes measured with the automated chambers. For $N_2O$ no real comparison can be made due to the malfunction of the $N_2O$ analyzer of the automated soil chamber set-up. Therefore the measurements of July and September from the automated chambers are compared to the measurements of August of the fast box chambers. Our results show that the fast box measurements are generally higher than the automated measurements. The mismatch of dates could lead to a discrepancy in the averages, especially because we see a slight increase in flux for some automated chambers in early August and then a decrease again at the end of August, which could indicate a period of higher fluxes that is missed here. This discrepancy could also be due to altered soil conditions  at the automated chamber locations due to the long term deployment on the same location"

[Figure]

*Figure 1: Flux measurements for $CO_2$, $CH_4$ and $N_2O$, with in red the measurements from the fast box chambers in plot CF1 and in black the measurements from all automated chambers.*

5. *N₂O emissions are highly related to soil nutrient availability, do you have information about the spatial variation of soil N among chamber locations? Table S2 shows the soil properties data but where the soils are sampled and how they can represent the different chamber measurement locations?*

 The article did not mentioned where the soil property data was measured. We have added to the caption of Table S2 that these measurements were made in the CF1 plot.
We do not have information on nutrient availability or soil N for all chambers separately. The CF1, CF2 and Mi5 plots are all dominated by Haplic Ferralsols and have a clay content of around 30% to 40% , while the area of Mi2, which is the plot furthest away from the others, situated slightly lower, is dominated by both Haplic and Xanthic Ferralsols and has a slightly smaller clay content, between 20%  and 30%. Mi2 is therefore slightly different from the other plots, but all soils are Ferralsols, kaolinitic, acidic with a pH in water less than 4.5, poor in organic carbon and in exchangeable cations. Therefore, only one set of soil parameters is included in the article. To give the reader more information about the homogeneity of the soils, we have included the following soil map in the Supplementary Material (Figure S10) and added a reference to a soil map, i.e.: "Gilson, P., Van Wambeke, A. and Gutzweiler, R.: Carte des Sols et de la Végétation du Congo Belge et du Ruanda-Urundi, 6: Yangambi, planchette 2: Yangambi. Notice explicative. INEAC, Bruxelles, 1956".

[Figure]

*Figure S10: Soil map of Yangambi with the CongoFlux climate site (0°48'52.0" N 24°30'08.9" E) in the black dotted line tower and the squares indication the locations of the 4 sampling plots (CF1, CF2, Mi2 and Mi5). Source for the map is: Gibson et al., 1956*

6. *Since the observed CH$_4$ emissions are not explained by moist soil or rain event, do you have more information can explain this? How about the ground vegetation in different plots, especially in the chamber location which show emissions throughout the period.*

   The forest, understory and ground vegetation are homogeneous in the four different plots. The four plots are completely inventoried and are comparable in species distribution. The chambers were randomly placed and in most of them ground cover vegetation was present, but due to the limited height of the chambers no mayor understory growth was possible. Visual inspection, did not reveal specific differences between the chambers in terms of vegetation and there was no clear difference between the times when a chamber was emitting or absorbing CH$_4$. Similarly for the automated chambers, there was no visual indication of whether a chamber was emitting or absorbing.

7. *Please check and clarify if the data comes from automatic chamber or fast box chamber in all table and figure captions.*

   Thank you for reading the captions thoroughly. I added in each caption if the shown metrics/measurements are either from the automated or the fast box chambers.

---

## Author Response (AR1)

**Author's response, iteration: Major revisions**

**EGUsphere-2024-2346**

*Editor review:*

*Dear Dr. Roxanne Daelman,*
*Thank you for providing a point-by-point answer to the comments made by the three reviewers. I invite you to submit a revised version of your manuscript that adequately deals with the points that were raised by the reviewers. Please provide a detailed description of the materials and methods to ensure the reproducibility of your study (e.g. location of manual vs. automatic chambers, time and number of measurements, data quality control, size of manual chambers vs. performance, integration of data from automatic vs. manual chambers), and identify / discuss (i) potential drivers of N2O emissions other than water-filled pore space (e.g. nutrient availability, climate drivers) and (ii) seasonal / interannual variability in N2O emissions.*

We thank the editor for the fast handling of our manuscript and we thank all the reviewers very much for their positive feedback and for their questions and comments. Below we list all questions and comments we received from the reviewers in black together with the answers and changes we made in the manuscript in blue. We grouped some questions related to the same topic together and highlighted per section the adaptations that we made in the manuscript. All line numbers refer to line numbers in the new revised document (without track changes), changes in the manuscript are in italics.

1. *The Congo Basin is a vast and diverse region. How do you ensure that the data collected from the specific study site in Yangambi is representative of the broader Congo Basin's tropical forest soils?*

   Thank you for your question. The Congo Basin is indeed a vast and diverse region with different climate, soil types, different forest compositions, rich biodiversity, and different forest types in general. The results of this study are therefore not representative of the whole Congo Basin. However, the CongoFlux site is situated in a lowland mixed species forest, identified as semi-deciduous with patches of evergreen forest. According to (Shapiro et al., 2021), semi-deciduous rainforest covers around 104 330 000 ha of the Congo Basin and a combination of evergreen and semi-deciduous forest covers a total area of 18 000 000 ha. In terms of vegetation, the CongoFlux site therefore represents about 33% of the entire Congo Basin, assuming 3.6 million square kilometer total size. Moreover, lowland semi-deciduous forests as found at our site represent 91% of all tropical forest types in the Congo Basin. The main soil type at our research site is Ferralsols. According to (Baert et al., 2009) Ferralsols are the dominant soil type in the DRC, which contains most of the tropical forest of the Congo Basin. We are therefore confident that our site is well-suited to represent a significant part of the tropical forest realm in the Congo Basin.

   The Congo basin in general lacks in situ-data and comparisons with the data that is available for soil fluxes show that there is quite a diversity of measurement techniques and results. With the combination of automated and fast box chamber we tried to tackle the problems of previous studies and therefore provide a more robust estimate. Although our

site is representative for a large area within the Congo Basin, this estimate will never cover the entire extend of the Congo Basin. However, it is a starting point for further investigation and a benchmark for model output.

We added in the Conclusion

*Line 401 -409: Despite being the second largest tropical forest worldwide, the Congo Basin is still generally understudied. Comparing the little available soil GHG flux data, shows that there is a diversity of measurement techniques and resulting GHG budgets. In this study, a combination of automated and manual fast box chamber measurements was used, to quantify and understand the spatio-temporal variability of soil GHG fluxes in a semi-deciduous tropical forest in the Congo Basin. The CongoFlux site is in terms of vegetation representative for around 33 % of the entire basin, assuming a total size of 3.6 million square kilometer (Shapiro et al., 2021). Moreover, the forest type found on the site represent 91 % of all forest types in the Basin. According to Baert et al., (2009), the main soil type of the CongoFlux site (Ferrasols) is also the dominant soil type in the DRC. We therefore believe that the results of this study provide a robust estimate, representative for a large area within the Congo Basin.*

Comments and questions relating to the comparison and integration of the two different methods

2. *You mention the use of both automated and manual soil chambers. Could you elaborate on how the data from these two different methods were integrated, and whether any corrections or normalizations were applied to ensure consistency in the dataset?*

   Thank you for bringing this to our attention. No corrections or normalizations were applied to either dataset. The automated chambers were installed to estimate robust annual fluxes which can only be achieved with high temporal resolution. The fast box method was included to reference the results of the automatic chambers into the larger study area. We believe that the methods are comparable due to the fact that the processing steps of the two methods are the same. When comparing the datasets, we take into account that the manual chambers are only measured during a limited period of time. The same time periods are selected to avoid comparing measurements in different meteorological situations. In the article, we have mentioned more clearly that these overlapping periods are selected for comparison.

3. *Still in relation to the fast boxes, when evaluating the performance of the chambers, it should be taken into account that with their small area, an increase in the variability of the fluxes was expected due to edge effects, while a larger area of the automatic chamber would reduce this influence on the fluxes.*

   The larger number of smaller chambers would indeed lead to an increase in variability compared to a smaller number of larger chambers. In this study we had to take the practicality and the feasibility of the method into account. We chose for the small and easy to handle chambers because the vegetation in the plots was dense, the trails were not easy to walk and the distances between plots were sometimes long. Larger chambers, to match

the size of the automated chambers would have been very difficult to manage with the small field team we had.

4. *Was there any place where the automatic chambers and fast box were placed close together in the same sampling location to evaluate the performances?*

5. *Since automatic chambers are at plot CF1, could you compare their results with fast box result in same plot to have an idea of the performance of the two methods?*

The automated chambers were located to the west, just outside the CF1 plot, so the manual fast box chambers of the CF1 plot were closest to the automated chambers. However, even the closest fast box chambers were already between 10 and 60 meters away from the automated set up. It is expected that $CH_4$ and $N_2O$ can already vary a lot between sites within only a few meters. Selecting the fast box chambers located at the CF1 plot, closest to the automated chambers, we can make the following comparison for $CO_2$, $CH_4$ and $N_2O$. The fast box measurements from the CF1 plot for $CO_2$ are quite comparable with those of the automated chambers over the same time period. The spread for the fast box method is a bit larger, which is to be expected since we have more locations and smaller chambers. In both the automated and the fast box measurements, positive fluxes for $CH_4$ are present, but the measurements are dominated by negative fluxes. Especially in the second half of August the fast box fluxes tended to be more negative than the fluxes measured with the automated chambers. For $N_2O$, no real comparison can be made due to the malfunction of the $N_2O$ analyzer of the automated soil chamber set-up during mid-August. Therefore the measurements of July and September from the automated chambers are compared to the measurements of August of the fast box chambers. Our results show that the fast box measurements are generally higher than the automated measurements. The mismatch of dates could lead to a discrepancy in the averages, especially because we see a slight increase in flux for some automated chambers in early August and then a decrease again at the end of August, which could indicate a period of higher fluxes that is missed here. This discrepancy could also to some extent be due to altered soil conditions at the automated chambers locations due to the long-term deployment on the same location, however no clear differences for the other GHG were detected in this period. We added in Supplementary material a comparison between the fast box and automated chambers located closest to each other.

We added in Section 2.3.2:

*Line 131 - 134: The low number of automated chambers limited the spatial coverage of the soil flux measurements. Hence, the fast box method was included to reference the results of the automatic chambers into the larger study area. The four GEM plots on the CongoFlux site were divided into twenty-five subplots of 20 m by 20 m and in each subplot, one soil chamber was installed in March 2023 to be measured with the fast box method (Hensen et al., 2013; Wangari et al., 2022).*

*Line 145 – 152: The quality control of these fluxes was similar to that of the automated fluxes, with the addition that fluxes with a low $R^2$ were also individually checked for their quality. Low $R^2$ could be due to a low flux and then the flux was put to 0. If the low $R^2$ was due to fluctuating*

*concentrations, the measurement was discarded. The size of the fast box chambers was smaller than the automated chambers and the number of chambers was larger, so the variation of the fluxes from the fast box chambers would likely be larger than that of the automated chambers. However, as the processing of the fast box data was the same as for the automated fluxes, we believe that the methods are compatible and we therefore can use the automated fluxes for budget calculation and the fast box measurements for referencing the spatial variability without any normalization or correction (Fig S11).*

*We added in supplementary material:*

*To evaluate the two methods used in this study, i.e. fast box and automated chamber method, a comparison between the measurements of the automated chambers and the measurements of the fast box chambers closest located to the automated chambers (plot CF1) was made during the overlapping time period. The fast box measurements from the CF1 plot for $CO_2$ were quite comparable with those of the automated chambers over the same time period (S11 a). The spread for the fast box method was larger, which is to be expected since there are more locations of the fast box chambers and the chambers are smaller in size. In both the automated and the fast box measurements, positive fluxes for $CH_4$ were present, but they were dominated by negative fluxes (S11 b). Especially in the second half of august the fast box fluxes tended to be more negative than the fluxes of the automated chambers. For $N_2O$ no real comparison could be made due to the malfunction of the $N_2O$ analyser of the automated soil chamber set-up. Therefore the measurements of July, only the beginning of August and September from the automated chambers were compared to the measurements of August of the fast box chambers (S11 c). Our results showed that the fast box measurements are generally higher than the automated measurements. The mismatch of dates could lead to a discrepancy in the averages, especially because we saw a slight increase in flux for some automated chambers in early August and then a decrease again at the end of August, which could indicate a period of higher fluxes that is missed here. This discrepancy could also be in some extend due to altered soil conditions at the automated chamber locations due to the long-term deployment on the same location, however no clear differences for the other GHG were detected in this period and the location of the chambers was consistently changed between to collars, so this effect should be minor.*

[Figure]

**Figure S11: Flux measurements for a) CO₂, b) CH₄ and c) N₂O, with in red the measurements from the fast box chambers in plot CF1 and in black the measurements from all automated chambers.**

Comments and questions relating to the fast box chamber method

6.  *Regarding fast boxes, is there any reference to their use in an experiment like the one presented or were they designed by the authors?*

    The fast measurements with the portable analyzer is briefly described in Hensen et al., (2013) and used in Bureau et al., (2017) and Wangari et al., (2022). The procedure in these articles is similar however the chambers are different across studies. We have added two references in the article in Line 134.

7.  *Line 130 described the manual chamber was permanently installed into soil, when they are installed and do you take into account any effect caused by the installation? Did you do quality control of fast box fluxes measurement as for automatic chamber? What's the percentage of bad quality measurements that are discarded?*

8.  *What are the intervals between samples for both types of chambers, the average number of measurements for the daytime and nighttime periods?*

9. *2.3.2: should provide detailed information about the experimental design for manual chamber measurements, which is crucial for understanding the methodology and interpreting the results. For example, clarify the time of day when measurements are taken and whether these times are consistent across all measurements or vary according to a specific schedule.*

The section about the fast box measurement indeed missed some crucial information for the reader to understand the method we applied correctly. We apologize for these unclarities and have now addressed these in the new version.

The installation of the manual chambers took place in March 2023. The chamber measurements used in this study were carried out in August 2023 and so the effects of installation can be discarded. The quality control performed for the fast box measurements was the same as for the automated chambers, except that measurements with low $R^2$, were also individually checked for their quality. Low $R^2$ could be due to a low flux and then the flux was put to 0. If the low $R^2$ was due to fluctuating concentrations, the measurement was discarded. Only 2 data points were removed and 5 were put to 0. The fluxes were measured manually with an analyser where the increase in greenhouse gas concentration could be seen visually on the user interface of the analyser. Bad seals or other problems were easily detected and the measurement was then restarted. Therefore, this small amount of poor quality data is as expected. For the automated a higher percentage of low quality data was removed. Low quality data was frequently due to the late detection of technical issues with the chambers or analysers.

For the fast box measurements, we added that only daytime measurements were available and that there was an average of 5 minutes between consecutive measurements, which was the time that it took to walk from one chamber to the next one. The route through all plots was changed every day such that the measurement timing of each chamber was different for every measurement.

We added in Section 2.3.2:

*Line 132 - 134: The four GEM plots on the CongoFlux site were divided into twenty-five subplots of 20 m by 20 m and in each subplot, one soil chamber was installed in March 2023 to be measured using the fast box method (Hensen et al., 2013; Wangari et al., 2022).*

*Line 138: During a period of three weeks from August 4 to August 28, 2023, flux measurements of $CO_2$, $CH_4$ and $N_2O$ were made during daytime (between 08:00 – 18:00) in these plots ....*

*Line 141-: Two plots were measured per day with an average of 5 minutes between consecutive chamber measurements. To avoid consistently measuring the same chamber on the same time of day, the order in which the plots were measured was alternated and the route from chamber to chamber within one plot was changed every session.*

*Line 145 – 152 The quality control of these fluxes was similar to that of the automated fluxes, with the addition that fluxes with a low $R^2$ were also individually checked for their quality. Low $R^2$ could be due to a low flux and then the flux was put to 0. If the low $R^2$ was due to fluctuating concentrations, the measurement was discarded. The size of the fast box chambers was smaller*

*than the automated chambers and the number of chambers was larger, so the variation of the fluxes from the fast box chambers would likely be larger than that of the automated chambers. However, as the processing of the fast box data was the same as for the automated fluxes, we believe that the methods are compatible and we therefore can use the automated fluxes for budget calculation and the fast box measurements for referencing the spatial variability without any normalization or correction (Fig S11).*

**Comments and questions relating to the automated chamber method**

10. *What are the intervals between samples for both types of chambers, the average number of measurements for the daytime and nighttime periods?*

11. *Please provide more details in 2.3.1 if the automatic chamber is opaque or not, this determine the measured $CO_2$ fluxes are NEE or respiration.*

    For the automated chambers, one out of nine chambers was closed every 17 minutes, resulting in one data point every 17 minutes, with almost the same number of day and night measurements due to the continued operation of the chambers. The automated chambers were opaque.

    We added in Section 2.3.1:

    *Line 104: At the CongoFlux site, nine custom-made dynamic automated chambers (Karlsruhe Institute of Technology) were installed just outside the 1 ha GEM plot CF1 (Fig. S2). The opaque chambers (0.5 m x 0.5 m x 0.15 m, length, width, and height) were controlled by a central steering unit consisting of a valve-tubing …*

    *Line 115: The chambers were installed in May 2022 and were operated with a closure time of fifteen minutes per flux measurement, followed by two minutes of purging with ambient air, resulting in one datapoint every seventeen minutes.*

    *Line 126: The data presented in this paper starts from the first of June 2022 until the 26th of September 2023, resulting in a coverage of sixteen months and 25 209 data points for $CO_2$ and $CH_4$ and 18 635 data points for $N_2O$ equally divided between nighttime and daytime measurements.*

**Comments and questions relating to the environmental variables and soil properties**

12. *It's not clear where the environmental variables are measured, is the "each chamber location" in Line 95 refers to which kind of chamber or both?*

13. *It is not clear in the text that the soil parameters shown in Table S2 were obtained only for the CongoFlux climate site and not for the other points of the experiment (CF1, CF2, Mi2 and Mi5). How representative are these measurements for the remaining points?*

14. *$N_2O$ emissions are highly related to soil nutrient availability, do you have information about the spatial variation of soil N among chamber locations? Table S2 shows the soil properties*

*data but where the soils are sampled and how they can represent the different chamber measurement locations?*

It was indeed not clear where the environmental variables were measured. We have added a sentence stating that these variables, i.e. VWC and soil temperature, are measured at the automated chamber locations and not at the fast box locations.

The article did also not mention where the soil property data was measured. We have added to the caption of Table 2 that these measurements were made in the CF1 plot.
We do not have information on nutrient availability or soil N for all chambers separately. The CF1, CF2 and Mi5 plots are all dominated by Haplic Ferralsols and have a clay content of around 30% to 40%, while the area of Mi2, which is the plot furthest away from the others, situated slightly lower, is dominated by both Haplic and Xanthic Ferralsols and has a slightly smaller clay content, between 20% and 30%. Mi2 is therefore slightly different from the other plots, but all soils are Ferralsols, kaolinitic, acidic with a pH in water less than 4.5, poor in organic carbon and in exchangeable cations. Therefore, only one set of soil parameters is included in the article. To give the reader more information about the homogeneity of the soils, we have included the following soil map in the Supplementary Material (Figure S10) and added a reference to a soil map, i.e.: "Gilson, P., Van Wambeke, A. and Gutzweiler, R.: Carte des Sols et de la Végétation du Congo Belge et du Ruanda-Urundi, 6: Yangambi, planchette 2: Yangambi. Notice explicative. INEAC, Bruxelles, 1956".

*In section 2.1 we added a reference to Figure S10*
*Line 81: The site is located in a semi-deciduous, lowland mixed-species forest with strongly weathered, sandy clay loam and poorly drained soils, dominated by Haplic Ferralsols (Gilson et al., 1956; Sibret et al., 2022, Fig S10).*

*In section 2.2 we added:*
*Line 95: Close to each automated chamber location, two water content reflectometers were installed …*

*Line 152: No additional climatic variables or soil properties were measured in the four GEM plots.*

*In the supplementary material we added Figure S10*

[Figure]

**Figure S10: Soil map of Yangambi with the CongoFlux climate site (0°48'52.0" N 24°30'08.9" E) in the black dotted line tower and the squares indication the locations of the 4 sampling plots (CF1, CF2, Mi2 and Mi5). Source for the map is: Gibson et al., 1956**

Comments and questions relating to the CH$_4$ fluxes

15. *The manuscript notes sporadic CH$_4$ emission events. What are the potential ecological or environmental triggers for these events, and how were they identified in your study?*

    Thank you for pointing out that this was not entirely clear. Line 343-356 says that several papers have suggested either heavy precipitation, termite activity or anoxic hotspots of microbial activity within the overall aerobic soil as possible triggers for these sporadic CH$_4$ emissions. In this study no clear evidence of termite activity was found at the chamber locations and the emissions did not occur only during wetter periods. Overall, this suggests that the emissions in our study are also associated with sporadically occurring anoxic microsites, dominated by methanogenesis.

16. *Since the observed CH$_4$ emissions are not explained by moist soil or rain event, do you have more information can explain this? How about the ground vegetation in different plots, especially in the chamber location which show emissions throughout the period.*

    The forest, understory and ground vegetation are homogeneous in the four different plots. The four plots are completely inventoried and are comparable in species distribution. The chambers were randomly placed and in most of them ground cover vegetation was present, but due to the limited height of the chambers no major understory growth was possible. Visual inspection, did not reveal specific differences between the chambers in terms of vegetation and there was no clear difference between the times when a chamber position

was a sink or a source for $CH_4$. Similarly for the automated chambers, there was no visual indication why or when a chamber was emitting or absorbing $CH_4$.

17. *Check if the $CH_4$ fluxes reported in lines 201 and 316 are correct, with the aforementioned tables.*

Thank you for checking the values in detail. The values on line 216 are correct. The range -133.1 to 1209.0 µg C m$^{-2}$ h$^{-1}$ can be found in Table S6, with chamber 4, collar 2 having the maximum value and chamber 5 collar one having the minimum value. The arithmetic mean in Line 217 (-44.6 ± 59.1 µg C m$^{-2}$ h$^{-1}$) is the mean of all measurements over all chambers and all collars. This mean value is not the same as the mean mentioned in the last row of Table S6 (-45.2 ± 21.8 µg C m$^{-2}$ h$^{-1}$). This last value represents the mean and standard deviation of all collar means.
The mean value of the fast box measurements in line 331 (-89.4 µg C m$^{-2}$ h$^{-1}$) is correct and can be found in Table 1. The range -230.8 to 256.99 µg C m$^{-2}$ h$^{-1}$ is the range of all measurements and not the average of the measurements per chambers, which are shown in Table 1.

The mean and range values of the automated chambers in line 333 (-66.8 µg C m$^{-2}$ h$^{-1}$ with a range of -162.8 to 272.2 µg C m$^{-2}$ h$^{-1}$) are calculated using the same period as the fast box measurements are carried out and are therefore not the same as mentioned in line 216.

*We added now small indications in the article to make these differences more clear. We added the reference to the table for the ranges and added the words 'calculated with all measurements' in the text to make clear that this is a different value from the means in Table S6. In the caption of Table S6, it is written that the "all chambers" row is calculated with the mean values per chamber, and not with all measurements combined.*

Comments and questions relating to the drivers of $N_2O$ fluxes

18. *The study identifies water-filled pore space (WFPS) as a significant driver of $N_2O$ emissions. Have you investigated other potential drivers, such as soil pH, nutrient availability, or microbial community composition, which could also influence $N_2O$ emissions?*

We thank the reviewer for this comment. Indeed having more data on other potential drivers would be interesting. But, we did not investigate other potential drivers apart from WFPS in this study. We do not have information on soil pH, nutrient availability or microbial community composition for all chambers separately. However, we state in line 395-396 that the low marginal $R^2$ of the fit suggests that there are indeed also other main drivers behind our results found in the study. We were primarily interested in capturing the high spatiotemporal variability using automated chambers to obtain a robust estimate for $N_2O$ flux magnitude and variability. Sampling for the mentioned other drivers even only at a weekly frequency was not feasible and not trivial in the setting of this study. Problems such as sample preservation hinder sensitive analysis such as microbial composition. With continued developments in the field (e.g. DNA preservation solutions) and improved infrastructure we hope to implement these parameters in future studies.

In Line 390 we mention that the change in microbial community composition could have been associated with the disappearance of the peaked responses of $N_2O$ to increasing WFPS

We *added in the result section 5.3:*

*Line 397: Other potential drivers could be soil pH, microbial community composition, nutrient availability, ... (Butterbach-Bahl et al., 2013) however in this study, we did not investigate these as sampling for the mentioned other drivers, even only at a weekly frequency, was not trivial and nor feasible.*

19. *The high temporal variability of $N_2O$ emissions is noted. Could the authors provide insights into the seasonal patterns and inter-annual variability observed in the study, and how this variability might be linked to climate drivers?*

In terms of seasonal variability and the drivers, we mention in the paragraph starting on Line 384 that we see higher emissions in the wet compared to the dry season. We also see higher emissions after rain events, but these seem to disappear with the onset of the drier season. Regarding the intra-annual variability, we mention that there is a large variability between consecutive years in the automated chamber measurements as we see lower fluxes during the onset of the second wet season around June and July than in the first wet season. However, during the second wet season, higher fluxes are measured by the manual chambers. which could indicate a period of higher fluxes that is missed in the automated flux measurements due to technical issues. This discrepancy could also be due to altered soil conditions at the automated chamber locations due to the long-term deployment on the same location. However no clear differences for the other GHG were detected in this period and the location of the chambers was consistently changed between two collars, so this effect should be minor.

*LINE 384 - 388: "WFPS is the strongest driver (Table S9) and has the largest relative effect size within its range of variability compared to other predictors (Fig. 2). The positive relationships fitted by the linear mixed model are found in several studies, confirming higher emissions during wet season, compared to dry seasons (Iddris et al., 2020; Werner et al., 2007). Shortly after rain events, $N_2O$ emissions increase rapidly and then slowly decrease again with decreasing WFPS (Fig. S9 c). From January 2023, with the onset of the drier months, the high fluxes and peaked responses to increasing WFPS seem to disappear."*

*LINE 373: "Emissions change significantly from year to year. The same months separated by only one year can differ in $N_2O$ emissions by a factor of four."*

*LINE 390 – 395: "With the onset of the early wet season around June and July, the emissions do not increase again. However, the fast box flux in August is almost the same as the high average flux measured by the automated chambers around the same period in the previous year (June and July 2022). The large difference could therefore also be the result of altered conditions at the chamber locations due to the long deployment of the automated chambers at the same location. However there was no clear difference for the other GHG fluxes during this period ant the location of the automated chambers was consistently changed between collars, so this effect should be minor."*

20. *What is the detection limit of the flux /GC? Are these real negative fluxes? please state detection limits - important when claiming negative fluxes.*

We thank the reviewer for noticing that the detection limits of the measurement instruments are not mentioned in the article. The negative $CH_4$ fluxes mentioned in the study are real in the sense that negative fluxes are generally expected for methane and $CH_4$ uptake has been measured before in other studies in similar areas, for example Barthel et al., 2022. The LICOR analyzers used in this study measure at a frequency of 1 Hertz. The linear fits are therefore performed on many datapoints which is in contrast to the static chamber method, analyzed with a GC, where linear fits are performed on 4 data points only. Therefore, we assume that the detection limits in our case are less important than when using the static chamber method linked to GC. We do know the precision of the analysers given by the company, from which we can deduce detection limits.

*We added to the supplementary material:*

**Precision (1σ) of the analysers used in the set-up**

**$CO_2$ Measurements**
*3.5 ppm at 400 ppm with 1 second averaging*
**$CH_4$ Measurements**
*0.60 ppb at 2 ppm with 1 second averaging*
**$N_2O$ Measurements**
*0.40 ppb at 330 ppb with 1 second averaging*

*Calculating the detection limit for the automated chambers by using 2 times the standard deviation divided by the closure time of 15 minutes as dq/dt in equation 1 from the article, results in a detection limit for $CO_2$ equal to 2.0 mg C $m^{-2}$ $h^{-1}$, for $CH_4$ equal to 0.3 µg C $m^{-2}$ $h^{-1}$ and for $N_2O$ equal to 0.5 µg N $m^{-2}$ $h^{-1}$.*

*Calculating the detection limit for the fast box chambers by using 2 times the standard deviation divided by the closure time of 2 minutes as dq/dt in equation 1 from the article, results in a detection limit for $CO_2$ equal to 15.5 mg C $m^{-2}$ $h^{-1}$, for $CH_4$ equal to 2.6 µg C $m^{-2}$ $h^{-1}$ and for $N_2O$ equal to 1.6 µg N $m^{-2}$ $h^{-1}$.*

21. *L120, give a reference for the equation used.*

We thank the reviewer for noticing that the use of this formula was not clear. The formula is a combination between the ideal gas law and additional scaling variables to arrive at the units we work with in this article. We reformulated the equation so that it is more clear how the equation is used.

*The equation is now written as:*

$$Flux = \frac{dq}{dt} \times \frac{V \times P \times M_w}{R \times T} \times \frac{60}{A \times 1000} \quad (2)$$

22. *Considering the occurrence of precipitation almost every day (Fig. S3), what is the strategy for measurements with these events?*

Rainfall was indeed a frequent event. During rain events, the automated chambers continued to measure as normal, without interruption. Originally, a longer chamber closure time was planned (45 minutes), with several chambers closing at the same time. A tipping bucket was installed in order to open all chambers during heavy rain events to ensure that the soil in the closed chambers did not miss out on an entire rain event and receive as much precipitation as the surrounding soil. We also changed the location from one collar to the other every two or three weeks for this purpose. The tipping bucket was not as sensitive as expected and did not trigger the opening on most rain events. However, we quickly shortened the closure time to 15 minutes, which reduced the chance of missing out on the whole rain event and only one chamber was closed at a time, which meant that only one chamber missed out on a part of the rain events. Therefore, we can assume that continuing the measurements during the rain events would not bias the data.

Manual measurements were continued during drizzle and light rain events, but were interrupted during too heavy rain events. Measurements were resumed as soon as possible after the rain event.

*We added in section 2.3.1*

*Line128:  During precipitation events, the automated chambers continued to measure without interruption.*

23. *In tab. 1, inform that the data refers to fast boxes.*

We have added this information

24. *Please check and clarify if the data comes from automatic chamber or fast box chamber in all table and figure captions.*

We thank the reviewer for reading the captions thoroughly. We added in each caption if shown metrics/measurements are either from the automated or the fast box chambers. We checked all figures and tables in the manuscripts and in the supplementary material and added the words "fast box" or "automated" to make the difference clear.